# Epigenetic Alterations of Repeated Relapses in Patient-matched Childhood Ependymomas

Sibo Zhao[1,2,3,4,18], Jia Li[5,6,7,18], Huiyuan Zhang[1,2,18], Lin Qi[1,2,8], Yuchen Du[1,2,8], Mari Kogiso[1,2], Frank K. Braun[1,2], Sophie Xiao[8], Yulun Huang[1,2,9], Jianfang Li[5], Wan-Yee Teo[10,11,12,13], Holly Lindsay[1,2], Patricia Baxter[2], Jack M. F. Su[2], Adekunle Adesina[14], Miklós Laczik[15], Paola Genevini[15], Anne-Clemence Veillard[15], Sol Schvartzman[15], Geoffrey Berguet[15], Shi-Rong Ding [16], Liping Du[17], Clifford Stephan[6], Jianhua Yang[2], Peter J. A. Davies [6], Xinyan Lu[17], Murali Chintagumpala[2], Donald William Parsons [2], Laszlo Perlaky[2], Yun-Fei Xia[16], Tsz-Kwong Man [2], Yun Huang [5], Deqiang Sun[5] ✉ & Xiao-Nan Li [1,2,8] ✉

Recurrence is frequent in pediatric ependymoma (EPN). Our longitudinal integrated analysis of 30 patient-matched repeated relapses ($3.67 \pm 1.76$ times) over 13 years ($5.8 \pm 3.8$) reveals stable molecular subtypes (RELA and PFA) and convergent DNA methylation reprogramming during serial relapses accompanied by increased orthotopic patient derived xenograft (PDX) (13/27) formation in the late recurrences. A set of differentially methylated CpGs (DMCs) and DNA methylation regions (DMRs) are found to persist in primary and relapse tumors (potential driver DMCs) and are acquired exclusively in the relapses (potential booster DMCs). Integrating with RNAseq reveals differentially expressed genes regulated by potential driver DMRs (*CACNA1H, SLC12A7, RARA* in RELA and *HSPB8, GMPR, ITGB4* in PFA) and potential booster DMRs (*PLEKHG1* in RELA and *NOTCH, EPHA2, SUFU, FOXJ1* in PFA tumors). DMCs predicators of relapse are also identified in the primary tumors. This study provides a high-resolution epigenetic roadmap of serial EPN relapses and 13 orthotopic PDX models to facilitate biological and preclinical studies.

Ependymoma (EPN) is the third most common malignant brain tumor of childhood, accounting for up to 12% of intracranial tumors in children. Current therapy includes maximal surgical resection and focal radiation, resulting in a 5-year overall survival (OS) and progression-free survival (PFS) of 70% and 57%, respectively[1–5]. However, nearly half of patients will experience late relapses[1,3,4]. Despite repeated treatment, most children with relapsed tumors eventually succumb to the disease. Ten-year OS and PFS decrease further to $50 \pm 5\%$ and $29 \pm 5\%$, respectively[1]. The use of chemotherapeutic agents for EPN has been extensively studied for decades, however, survival benefit remains

controversial[6,7]. In-depth understanding of the biology of EPN recurrence is needed.

While it is well established that tumor location is important in EPN biology[8], recent studies have shown that epigenetic changes mediated by DNA methylation play an important role in EPN tumorigenesis[9–11] as gene mutations in EPN are much less frequent than in adult cancers[9–11]. Indeed, non-mutational epigenetic reprogramming has recently been incorporated as an emerging hallmark of cancer[12]. Similar to several other types of pediatric cancers[13–15], DNA methylation analysis has successfully subclassified pediatric EPNs into nine molecularly subgroups

---

with distinct clinical outcomes[16,17]. Although oncogenic drivers have been identified for primary EPN[8,11,15,16,18-23], understanding of recurrent EPN biology is still at its infancy. One of the challenges is the difficulties of obtaining relapsed tumor tissues. Although longitudinal analysis of consecutive, serially relapsing patient tumor samples will enable the separation of serially conserved potential epigenetic driver(s) from transient alterations, it remains very difficult and requires a committed collaborative team of physicians, neuropathologist and tumor biologists. Using single recurrent EPN tumor with or without patient-matched primary tumors is a frequent approach. The second challenge lies in the reliable detection of subtle genetic/epigenetic changes. As a recent study revealed that morphological changes in recurrent EPNs was not sufficiently explained by epigenetic changes detected with 450k DNA methylation array[24], improved DNA methylation site coverage combined with integrated analysis of gene expression patterns[9,10,25-28] should facilitate the discovery of genetic interactions and new therapeutic opportunities.

A third challenge in studying EPN relapse is the limited availability of animal models. After the establishment of the first supratentorial EPN patient-derived orthotopic xenograft (PDOX, or orthotopic PDX) model by our group[29], we and others have shown that direct implantation of patient tumors into the matching locations in mouse brains promoted the establishment of clinically relevant animal models that replicate histopathological features, invasive/metastasis phenotype and genetic profiles of the original patients[29-35]. While several genetically engineered animal models have been developed[22,23,36], correlating tumorigenicity of patient EPN tumors during tumor progression, i.e., from diagnosis till consecutive, serial relapses of each patient, should serve as an in vivo functional assay to determine key tumorigenic drivers and identify druggable targets for EPN relapses.

Here, we report a deep longitudinal analyses of DNA methylation landscape using single-base-resolution DNA methylome in a cohort of 30 patient-matched primary and serially relapsing EPN tumors together with integrated analysis of RNA-seq profiles and functional examination of PDOX tumorigenicity for this set of matched primary and relapsed EPNs. Our goal is to apply this integrated analyses and clinically relevant patient-matched PDOX mouse models from the same cohort of patients to identify the potential drivers of relapse (abnormal DNA methylation present in primary tumor and sustained in all relapsed EPNs), the potential boosters of relapse (DNA methylation newly acquired in relapsed tumors and persisted with tumor progression), and potential predictors of relapse in the primary tumors (DNA methylation in the primary tumors which predicts future/subsequent relapse).

## Results
### Maintenance of molecular subtypes during serial relapses of EPN
The distinct molecular subtypes of pediatric EPN at diagnosis can impact clinical treatment[13,14,16,37,38]. While analysis of single recurrences suggested the maintenance of molecular subtypes, it remains to be determined if the molecular subtypes of EPNs will change over repeated relapses of a long period of time. Despite 50% EPN patients develop tumor recurrences[1,3,4], relapsed tumor samples, particularly the repeated recurrences, are difficult to obtain. In our study, we collected a total of 30 serially relapsing EPN tumor samples from 10 of 110 (9.1%) pediatric patients who were followed up over 13 years. Consent to publish clinical information potentially identifying individuals was obtained. Many recurrent tumor samples were obtained after the patients were treated with chemo- and/or radiation therapies (Fig. 1A). The time from diagnosis to last recurrence ranged from 2.75 to 13 years, and the number of recurrences per patient ranged from 1 to 7 times (3.67 ± 1.76 times) (Fig. 1A, Table 1).

To achieve high-resolution analysis of DNA methylation, we performed genome-wide DNA methylation sequencing using four normal pediatric brain tissues (2 cerebellar and 2 cerebral tissues) (Table 1) procured from warm (<6 hr) autopsy of two children as controls (Fig. 1A, Table 1). In all the samples, we identified >2 × 10⁶ CpGs each covered by at least five reads with a mean DNA methylation ratio at 51% (Supplementary Data 1). The EPN tumors exhibited DNA methylation profiles distinct from the normal tissues (Supplementary Fig. 1A, B). Similar to their primary tumors, the recurrent EPNs were subclassified into 5 RELA and 5 PFA tumors that displayed clear segregation in phylogenetic construction using the top 6000 variant CpGs (Fig. 1D, E and Supplementary Fig. 1C, D) as well as the maintenance of key DNA methylation signatures (Fig. 1C) and gene expression patterns that were previously identified in primary EPN tumors (GSE64415)[39] (Fig. 1F, Supplementary Data 2). These set of data demonstrated the maintenance of EPN molecular subtypes during repeated relapses (≥2) during years of chemo- and/or radiation therapies.

Additional analysis of CpG islands detected an increase of DNA methylation levels in the relapsed tumors compared to the primary EPNs, of which the levels of CpG islands were higher than the normal brain tissues (Supplementary Fig. 1E). In contrast, no consistent DNA methylation changes on CpG shore regions (flanks CpG islands up to 2 kb away from the CpG islands) were identified in the relapsed tumors (Supplementary Fig. 1E).

Non-CpG methylation is recently recognized as a layer of epigenetic information assembled at the root of vertebrates and plays new regulatory roles independent of the ancestral form of the CpG methylation[40]. Its patterns are often tissue-spcific[40-43]. In the present study, the primary and recurrent tumors were found to have a dramatically decreased mCpA levels in both RELA and PFA tumors (Fig. 1G) when compared with the normal tissues. Although the differences between RELA and PFA tumors were not significant, this finding suggested that the reduction of mCpA can be a potentially epigenetic signature of EPN tumor of which the functional roles warrant further examination.

### Convergence of DNA methylation landscape during repeated relapses
It remains unknown how EPN progression and repeated relapses affect cellular subpopulations. To understand the impact of selective, adaptive and progressive pressures on epigenetic reprogramming during long-term serial relapses of EPN, we calculated the Pearson correlation coefficient of DNA methylation ratios between the consecutively relapsing tumors during tumor relapses for each patient. In the event an adjacent previous relapsing tumor was missing, a previously available sample was used to calculate the correlation. Unlike the low-level correlations between primary tumor and normal brain tissues (average $r$ of 0.24 for RELA and 0.23 for PFA), the Pearson correlation between two longitudinal consecutively relapsing tumors increased significantly, reaching 0.75 in 8/11 pair-wise comparisons in RELA recurrences, and 0.83 in 6/7 PFA recurrences (Fig. 2A, Supplementary Fig. 2A–C), respectively. For example, in RELA1 the correlation coefficient ($r$) increased to 0.68 between the 3rd and 2nd recurrence, 0.70 between the 4th/3rd recurrence, and 0.77 between the 7th/5th recurrence, demonstrating a trend of convergence during repeated recurrences (recurrent times ≥ 2). A similar pattern was also exemplified for PFA1 (Fig. 2B). Parallel analysis of transcriptional data detected a similar trend of increased correlation (Supplementary Fig. 2D) as well. As no previous samples/studies were available, our data provided more insight on EPN relapse. This finding is also important, as the epigenetically homogeneous tumors can theoretically be more effectively targeted than the widely heterogeneous tumors. Unlike other cancers exhibiting mutational divergent at relapses[44], this result indicated that pediatric EPNs harbor epigenetic lesions that are selectively enriched by clinical treatments (chemo- and/or radio-therapies) and biological evolution in driving the progression of tumor recurrences.

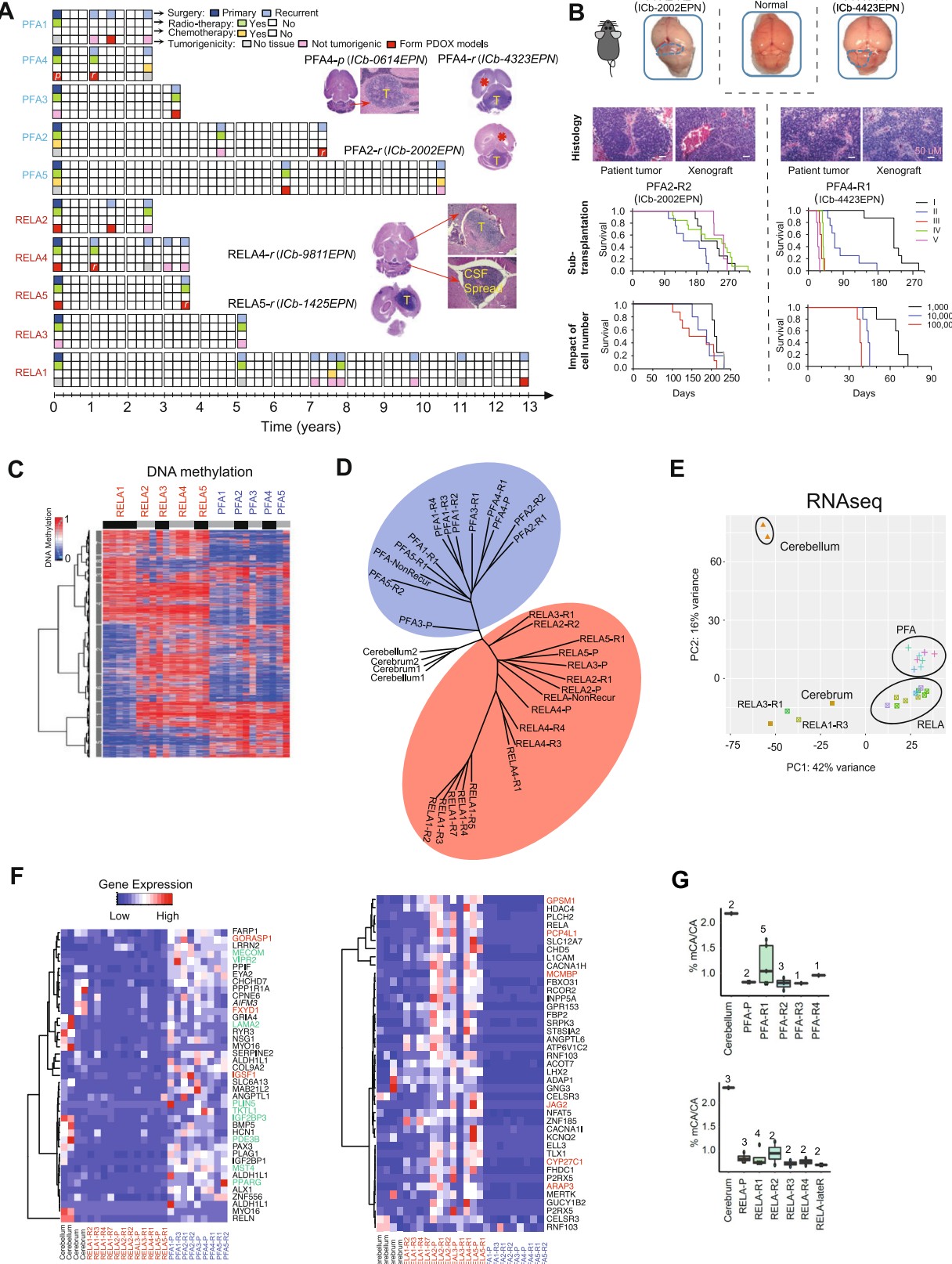

## Late EPN relapses exhibited increase of tumorigenicity in the brains of SCID mice

We next investigated if the progressive and convergent epigenetic changes of the recurrent tumors are functionally important in promoting increased tumorigenic capacities. Despite the success in developing PDOX models for pediatric malignant brain tumors[29,31,32,34,35,45–49] by our group and others, childhood EPN is notoriously known for low tumor take rate[29,50,51]. With our discovery of the progressive and convergent nature of epigenetic changes during serial EPN relapses, we hypothesized that such epigenetic reprogramming enabled the recurrent tumors to acquire tumorigenic capabilities. Therefore, we systematically implanted 27 EPN

**Fig. 1 | DNA methylation and transcriptomic landscape of recurrent ependymoma tumors. A** Time-course summary of surgery, radio- and chemotherapy treatment and tumorigenicity information of the 10 sets of recurrent ependymomas. Images of intra-cerebellar (**ICb**) and intra-cerebral (**IC**) orthotopic xenograft tumor formation from primary (**p**) or recurrent (**r**) PFA and RELA tumors (model ID in round brackets) were inserted to show tumor (**T**), hydrocephalus (*) and CSF spread. **B** Orthotopic xenograft mouse models of recurrent PFA EPNs. Tumor formation can be seen on gross (top) mouse brains (outlined). H&E staining (middle) showing histological comparison between the originating patient tumor and the PDOX tumors. Changes of animal survival times (lower panels) during serial in vivo transplanations of intra-cerebellar (*ICb*) PDOX models from passage I (*I*) to V (*V*) and the impact of the implanted different cell numbers (from 1000 cells per mouse to 100,000 cells) on animal survival times of two PDOX models were shown (*n* = 10 mice/group). Scale bar = 50 μM. **C** Heatmap showing the DNA methylation ratios of the top variable 20,000 CpGs across all EPN tumor samples. The patient ID is labeled vertically on the top of the heatmap. The black and gray bars under the patient IDs are used to separate tumors from the same patient. **D** Phylogeny tree

construction using the top 6000 CpGs with variable DNA methylation ratios in RELA (red circle) and PFA (blue circle) tumors. **E** Principle component analysis of RNA-seq data from EPN tumor samples using age matched childhood normal brain tissues as references. **F** Heatmap representing the PFA (*left*) and RELA (*right*) primary tumor signature genes' expression levels in all samples, including multiple relapses. The primary signature genes were selected from a previously published database GSE64415[16,39,88], and those overlapped with our consistently decreased (green) and increased (red) genes were highlighted. **G** CA methylation ratios in PFA (left) and RELA (lower) primary and recurrent tumors as compared with normal childhood cerebellar and cerebral tissues. Boxplots indicate median, first and third quartiles (Q1 and Q3), whiskers extend to the furthest values; the uppermost and lowest line indicates the maximum and minimum values, respectively. Marked on the top of each boxplot is the number of samples analyzed including Cerebellum (*n* = 2); Cerebrum (*n* = 3); PFA-P (*n* = 2); PFA-R1 (*n* = 5); PFA-R2 (*n* = 3); PFA-R3 (*n* = 1); PFA-R4 (*n* = 1); RELA-P (*n* = 3); RELA-R1 (*n* = 4); RELA-R2 (*n* = 2); RELA-R3 (*n* = 2); RELA-R4 (*n* = 2); RELA-lateR (*n* = 2).

**Table 1 | Summary of clinical information of the ependymoma patients and the autopsied normal tissues**

| Patient ID (Dx) | Age (years) | Gender | Tumor location (primary and recurrent) | Total number of recurrences | Time to first recurrence | Time to last recur | Patient status |
|---|---|---|---|---|---|---|---|
| **Recurrent tumors** | | | | | | | |
| PFA1 | 3 | Female | Posterior fossa | 3 | 14.7 months | 35.8 months | Alive |
| PFA2 | 7 | Male | Posterior fossa | 2 | 56.9 months | 90.3 months | LTF |
| PFA3 | 2 | Male | Posterior fossa | 1 | 43.6 months | N/A | Alive |
| PFA4 | 4 | Male | Posterior fossa | 2 | 13 months | 33.2 months | Alive |
| PFA5 | 8 | Female | Posterior fossa | 3 | 60.9 months | 131.5 months | LTF |
| RELA1 | 8 | Female | Right fronto-parietal | 7 | 60 months | 170.4 months | Deceased |
| RELA2 | 10 | Male | Right frontal | 2 | 22.2 months | 32.2 months | LTF |
| RELA3 | 7 | Male | Right frontal | 1 | 62.7 months | N/A | Deceased |
| RELA4 | 2 | Male | Left frontal | 4 | 12.4 months | 44.3 months | Deceased |
| RELA5 | 6 | Male | Right parietal | 1 | 45 months | N/A | Deceased |
| **Patient ID** | **Age (years)** | **Gender** | **Tumor location** | **Total recur number** | **Total follow-up duration** | | **Patient status** |
| **Non-recurrent tumors** | | | | | | | |
| PFA6 | 2 | Male | Posterior fossa | 0 | 131 months | | Alive |
| RELA6 | 6 | Male | Right frontal | 0 | 89 months | | LTF |
| **Tissue ID** | **Age (years)** | | **Gender** | | **Tissue location** | | **Source** |
| **Normal Brain Tissues** | | | | | | | |
| A1429-NC | 9 | | Male | | Right frontal | | Autopsied |
| A1429-NCb | 9 | | Male | | Right cerebellar | | Autopsied |
| A1123-NC | 5 | | Male | | Right frontal-parietal | | Autopsied |
| A1123-NCb | 5 | | Male | | Right cerebellar | | Autopsied |

*Dx* diagnosis, *Recur* recurrence, *NC* normal cerebrum, *NCb* normal cerebellum, *LTF* lost to follow-up.

tumors (6 primary and 21 recurrent) into the matching locations in the brains of SCID mice following our standardized protocols (same locations, same depth, both sexes of animals of similar age)[29–31,34,52] (Fig. 1A). All the mice received the same number ($1 \times 10^5$) of viable tumor cells. The animals were closely monitored for up to 15 months and euthanized if they develop neurological deficits or became moribund. Formation of intra-cerebral (IC) or intra-cerebellar (ICb) xenografts was confirmed via gross and histological examinations in 13 patient tumors (48.1%). Among the 5 patients whose primary tumors (*n* = 1 patient: PFA3) or early recurrent tumors (*n* = 4 patients: RELA1, PFA5, PFA2, PFA1) did not form xenografts, their late recurrent tumor(s) formed PDOX tumors. In RELA1, tumor formation was confirmed in the 7th recurrence after repeated failure of tumor growth at 2nd, 3rd, 4th, and 5th relapse (no patient tumor tissues were available at 1st and 6th recurrence). In PFA1 and PFA2 (Fig. 1A), PDOX formation was confirmed in the 2nd recurrence. Although tumorigenicity can be affected by many factors, our unique approach of testing serially relapsed tumor tissues from the same patients and using the standardized tumor implantation protocol

provided functional data to support the malignant progression of tumor recurrence and suggested a role of the convergent epigenetic reprogramming in promoting tumorigenicity in EPN relapses.

Many of the models have been sub-transplanted in vivo in mouse brains for up to five passages while exhibiting a reverse correlation between tumor cells implanted (1000–100,000 cells/mouse) and animal survival times (Fig. 1B). This set of models will provide a much-needed platform to translate biological findings to functional and preclinical testing for relapsed EPNs. To determine their molecular fidelity, we compared the global mean DNA methylation ratios and DMRs in 7 pairs of patient-xenograft tumors and detected a high-level of similarities ($r > 0.9$) in 6/7 models (Fig. 2D, E, Supplementary Fig. 2A–D) even during subtransplantations (RELA5 and PFA4), demonstrating the faithful recapitulation of patient epigenetic profiles in their matched PDOX models (Fig. 2E).

Gain of chromosome 1q has been associated with recurrence and poor prognosis in PFA tumor[1,50,53]. To examine if 1q gains might

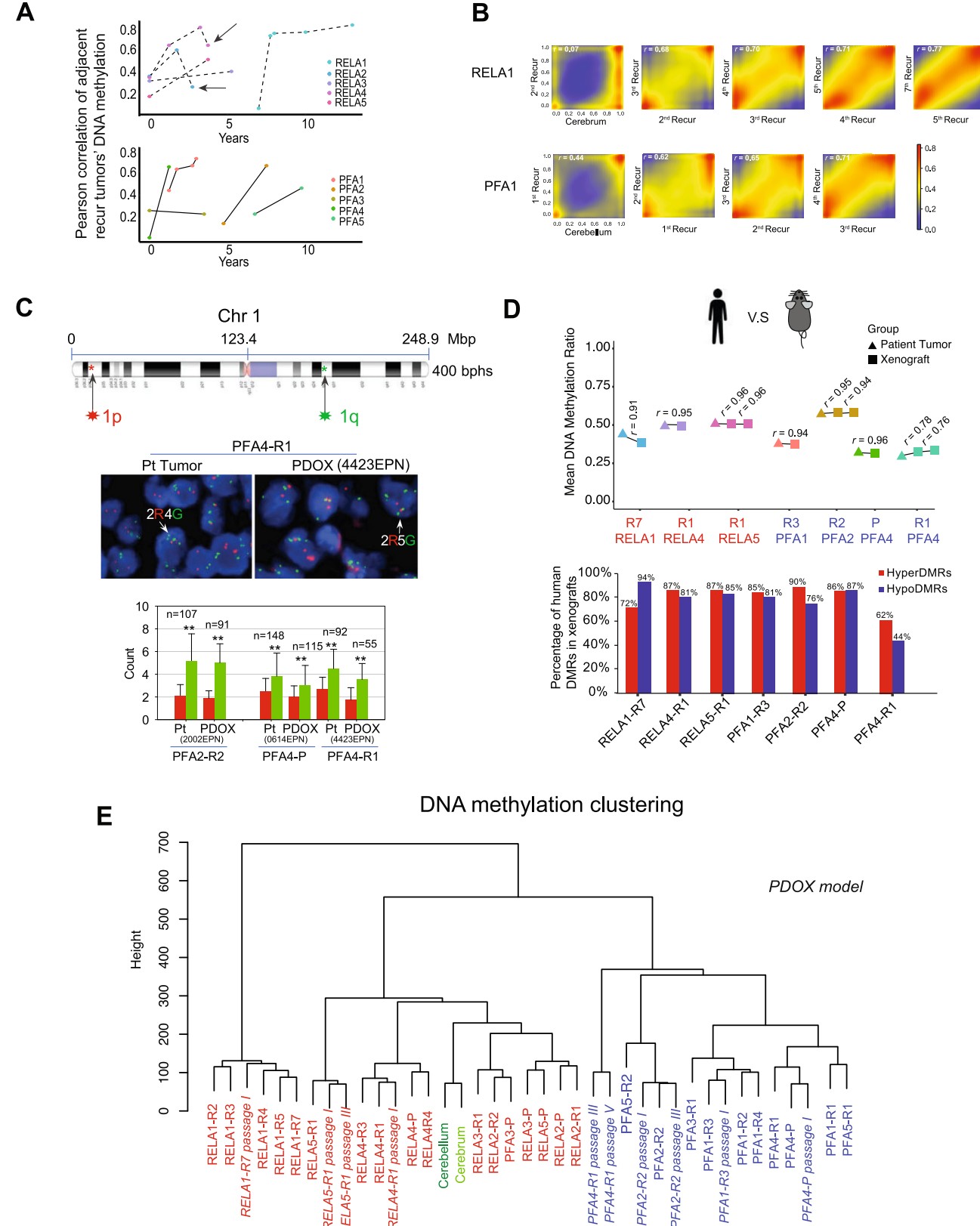

also be associated with tumorigenicity, we applied CNVkit to infer CNV status using our RRBS data. Due to the technique limitations, i.e., the identical reads and uneven coverage in tumors resulted from the enzyme digestion, there was not definitive identification of 1q gain in PFA patient and PDOX tumors (Supplementary Fig. 3). To supplement this approach, we applied FISH and detected 1q gain in

paraffin sections of three sets of patient and xenograft tumors of PFA ependymoma (Fig. 2C), suggesting that 1q gain in patient tumors were preserved in the PDOX tumors as well. These data support the analysis of additional PFA tumors, both tumorigenic and non-tumorigenic, to establish the role of 1q gain in PFA tumorigenicity.

**Fig. 2 | Progressive convergence of DNA methylation profiles during repeated ependymoma relapses. A** Line charts showing the changes of DNA methylation correlations between adjacent recurrent tumors during repeated recurrences of RELA (upper panel) and PFA (lower panel) tumors. Each dot represents one tumor sample and graphed to the time of recurrence. **B** Representative smoothed density scatterplots displaying the increased correlation coefficient (*r*) of DNA methylation profiles between the adjacent recurrent tumors of RELA1 (upper panel) and PFA1 (lower panel) patients. **C** FISH analysis of chromosome 1q gain showing the locations of FISH probes in 1p (red) and 1q (green) (top), representative images of 1q (*G*: green) gain relative to 1p (*R*: red) (middle) in matching pairs of patients (*Pt*) and PDOX tumors. The number of cells counted (*n*) was marked on top of the column of every sample (red bar indicates 1p count, and green bar indicates 1q count). Statistical analysis was performed through two-sided Student *t*-test. \*\**P* < 0.01. *P*-values = 7.7186E-30 (Pt-2002), 5.32303E-34 (ICb-2002EPN), 1.17953E-13 (Pt-0614), 1.5575E-08 (ICb-0614EPN), 4.53091E-21 (Pt-4423), 5.4252E-14 (ICb-4423EPN). Data are presented as mean values ± SD. (Magnification:×100). **D** Preservation of DNA methylation profiles of patient tumors in their matching PDOX models either from the primary (*P*) or recurrent tumors from the first (*R1*) up to the 7th (*R7*) recurrences. Global Pearson correlation (*r*) of the DNA methylation profiles from the matched patient tumor and PDOX tumors were labeled above the connected line. The numbers of CpGs used in the correlation analysis were 2,211,714 (RELA1-R7), 2,156,125 (RELA4-R1), 2,075,950 (RELA5-R1), 1,994,148 (PFA1-R3), 2,473,879 (PFA2-R2), 1,728,999 (PFA4-P), and 1,200,936 (PFA4-R1). Majority of the patient DMRs were maintained in their matching PDOX tumors (lower panel). The total numbers of hyperDMRs that were used in the analysis in patient and PDOX tumors are 6919 and 6265 (RELA1-R7), 10,868 and 12,825 (RELA4-R1), 11,406 and 14,333 (RELA-R1)m 5815 and 6653 (PFA-R3), 13,541 and 15,036 (PFA2-R2), 4107 and 8309 (RFA4-P), 3155 and 8373 (RFA4-R1); whereas the total numbers of hypoDMRs were 15,147 and 27,594 (RELA1-R7), 5797 and 8506 (RELA4-R1), 4361 and 7009 (RELA-R1), 12,051 and 15,519 (PFA-R3), 5824 and 5847 (PFA2-R2), 5373 and 10,917 (RFA4-P), 8969 and 10,050 (RFA4-R1), respectively. **E** Unsupervised clustering of primary/recurrent tumors from the first (-R1) up to the 7th (-R7) relapse as well as matching PDOX tumors at specific passages (passage).

## Potential DNA methylation drivers of EPN recurrence identified

To determine the fate of abnormal DNA methylation in the primary tumors and identify potential epigenetic drivers of recurrences, we performed a deep longitudinal analysis of the repeated recurrences to segregate the differential methylated sites (DMCs) consistently present during serial relapses from random DMCs in-transit (Fig. 3A, Supplementary Data 3). We used alluvia plot to show the dynamic changes of DNA methylation in the repeated relapses (≥2). Most of the DMCs underwent single direction changes while less than 2% of DMCs switched between HyperDMCs and HypoDMCs (Fig. 3B, Supplementary Fig. 4A). When the tumors were analyzed individually, they exhibited wide-ranges of variabilities, i.e., 28,791 to 99,702 (54,380 ± 24,495) HyperDMC and 13,662 to 84,714 (44,672 ± 21,560) HypoDMC in RELA and PFA recurrences. To identify the DMCs shared by >2 patients (Fig. 3C, vertical histogram) or maintained in each patient during multiple relapses (Fig. 3C, horizontal histogram), we applied UpsetR to plot the numbers and identified 8155 HyperDMCs and 2324 HypoDMCs shared by all the five RELA patients (HyperDMC$^{Shared}$ and HypoDMC$^{Shared}$) (Fig. 3C); and 6845 HyperDMC$^{shared}$ and 6925 HypoDMC$^{shared}$ shared by the five PFA patients. Many of these shared DMCs, ranging from 61.4 to 83.8%, were located in proximal regulatory regions of genes. To further increase the stringency of potential DNA methylation driver discovery, we focused on the DMCs that exhibited DNA methylation ratio differences greater than >0.3 between tumor and normal cerebellum/cerebrum tissues. From the DMC$^{shared}$ we identified 57 consistent-HyperDMC$^{shared}$ and 51 consistent-HypoDMC$^{shared}$ in RELA tumors; and 148 and 118 in PFA tumors, respectively (Fig. 3D, Supplementary Data 4). These DMCs persisted from the primary tumors to late relapses, thereby constituting potential DNA methylation driver signatures of EPN relapse.

## Potential DNA methylation drivers regulated a set of differentially expressed genes

To identify the target genes of the potential driver DMCs, we examined the differentially methylated regions (DMRs), which are groups of neighboring DMCs that play a more important role in regulating gene expressions than the single DMCs. In RELA primary and recurrent tumors, we detected an average of 9730 HyperDMRs and 8802 HypoDMRs, slightly higher than the 7910 HyperDMRs and 7977 HypoDMRs in PFA tumors (Supplementary Data 5) that were distributed on different functional elements (Supplementary Fig. 4B). Recognizing the impact of locations on the biology, we compared the primary and recurrent tumors between RELA and PFA EPNs (Supplementary Fig. 5) and identified 2131 and 976 consistent Hyper/HypoDMAs presented only in all recurrent samples. Cross examination with a public dataset (GSE65362)[16] revealed a high-level similarity of DNA methylation ratios of the CpGs within our DMRs (Supplementary Fig. 5B).

DNA methylation in promoters can repress gene expression[54]. To identify functionally important DMRs, i.e., those associated with genes, we focused on the most significant DMRs that were located within gene transcription start sites (TSS) up/downstream 5 kb of known target genes (Supplementary Fig. 4C, D). In the RELA patients, 588 potential driver genes (529 associated with the Hyper-DMRs and 59 with the Hypo-DMRs) were shared by all the patients (Supplementary Data 6); whereas in PFA patients, there were 736 potential driver genes (522 associated with the Hyper-DMRs and 214 with the Hypo-DMRs) (Supplementary Data 7). To further identify potential DMR drivers that were required for the PDOX tumor formation, we filtered potential patient DMR drivers with PDOX DMRs. In RELA patient tumors, 90.5% (479/529) of hyperDMR- and 89.8% (53/59) of hypoDMR-associated potential driver genes were preserved in their PDOX models, whereas in PFA tumors, it was 95.4% (522/547) and 93.8% (214/228), respectively (Fig. 3E, Supplementary Data 8). This integrated analysis of patient tumors with their matching PDOX tumors represent a strategy for the discovery of functionally important target genes of potential DMR drivers.

To determine if the levels of these target gene expressions were actually regulated by the potential driver DMRs in EPN relapses, we analyzed the RNA-seq data from the same batch of samples to detect the differentially expressed genes (DEGs) (Supplementary Fig. 6A) that were significantly different (*FDR* < 0.05) from the normal cerebella or cerebra, followed by matching the DEGs with the potential DMR drivers that displayed average differences of DMR ratios between cerebella/cerebra and tumors >0.3. Comparison between PFA and RELA tumors revealed that the PFA tumors shared more DEGs (upregulated = 1303 and down = 1290) and by three groups of tumors (from PFA-P to PFA-R1 and PFA-R20, which were remarkably higher than that in RELA tumors (up = relapses by three groups of tumors from PFA-P to PFA-R1 and PFA-R2 were significantly higher than those in RELA tumors (upregulated = 208 and downregulated = 281) that were selectively shared by RELA-P and RELA1 only (Supplementary Fig. 6A). The differences of cell-of-origin between RELA and PFA tumors[20–23,38,55] may have contributed to the differences of DEG panels and the relatively conserved DEGs during PFA relapses.

In RELA tumors, we identified 34/38 (89.5%) downregulated target genes (including *SYN2, PHACTR3, KCNJ9, RIMS4, FBOX41*) of the consistent-hyperDMRs$^{shared}$, and 5/9 (55.6%) upregulated target genes (*CACNA1H, SLC12A7, CSPG4, RARA,* and *ZNF423*) of the consistent-HypoDMRs$^{shared}$ (Fig. 3F). These genes accounted for 2.4% of the 208 upregulated genes and 12.1% of the 281 downregulated genes in the RELA tumors. Similarly, in PFA tumors, 81/151 (53.6%) downregulated genes (including *FAT1, MYT1L, GRM4, KCNK9, PCDHA5,* and *KCNA1*) were regulated by the consistent-hyperDMRs$^{shared}$ and 54/76 (71.1%) upregulated genes (including *FAM92B, HSPB8, GMPR, ITGB4, FHAD1,* and *FXYD1*) by the consistent-hypoDMRs$^{shared}$ (Fig. 3F, Supplementary

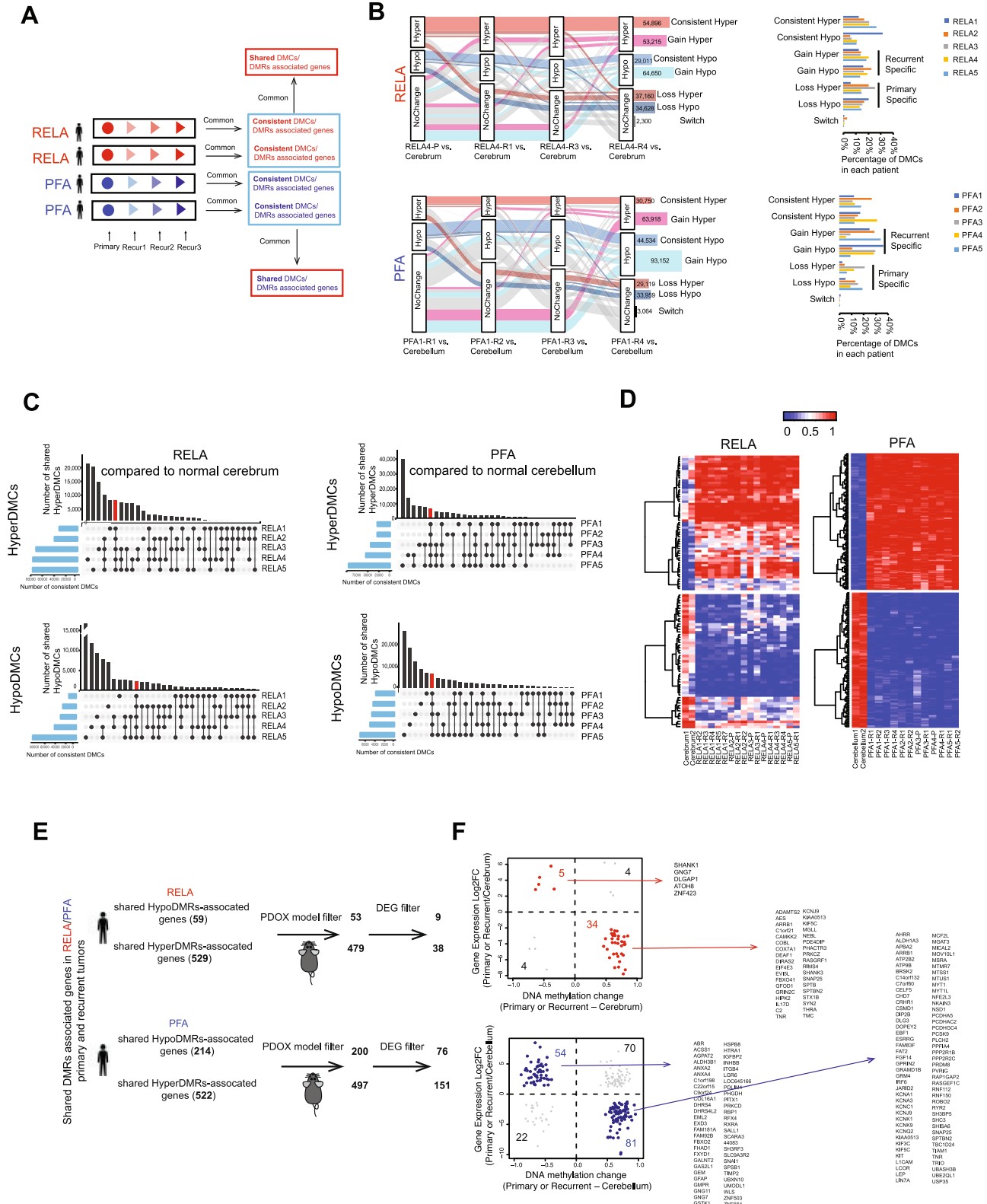

Data 5). They accounted for 4.1% of 1303 upregulated genes and 6.3% of the 1290 downregulated genes in the PFA tumors (Supplementary Fig. 6A). Although the differences of cell-of-origin between RELA and PFA tumors[20–23,38,55] may have contributed to the differences of DEG panels, the numbers of the shared upregulated ($n = 1303$) and downregulated ($n = 1290$) DEGs during PFA relapses from PFA-P to PFA-R1 and PFA-R2) were significantly higher than those in RELA tumors Functionally, these genes were associated with neuron development,

neuron differentiation and neurogenesis (Supplementary Fig. 6B). The strong negative correlations between the DMRs and their target genes (Fig. 3F) indicated a potential regulatory role of the potential DMR drivers in regulating the gene expressions. Most of the aforementioned DMR regulated genes were not previously discovered for EPN relapses. Their potential as relapse driver genes was further enhanced by the fact that many of them, including *CACNA1H*[56], *SLC12A7*[57], *CSPG4*[58,59], *RARA*[60] in RELA, *and HSPB8*[61], *ITGB4*[62], *FAT1*[63,64] in PFA

**Fig. 3 | Identification of potential DNA methylation drivers for ependymoma relapse. A** Schematic illustration of dynamic analysis of time-course DNA methylation to identify consistent DMCs that were present from primary to all recurrent tumors in each individual patient, and the shared DMCs that were consistently present in all RELA or PFA patients. **B** Representative alluvia plots (*left panel*) showing the dynamic changes of DNA methylation for RELA4 (left upper panel) and PFA1 (left lower panel) tumors during repeated recurrences. Normal brain tissues were used as reference to determine the CpG status, i.e., Hyper-, Hypo-methylation and No Change. Seven different patterns (categories) and the numbers of CpG changes were listed with different colors. Graphs (right panel) showed percentages of the 7 different categories of CpG changes of each RELA (right upper panel) and PFA patient (right lower panel). **C** UpSet R plots showing the number of consistent Hyper- and HypoDMCs (*y*-axis) that were shared among RELA (left panels) and PFA patients (right panels) (connected dots with lines) as well as numbers of consistent DMCs for each patient (the horizonal histograms). Consistent DMCs that were shared by all RELA or PFA accounted for a small fraction and were highlighted in red, respectively. **D** Heatmaps showing the DNA hyper- (red) and hypo-methylation (blue) ratios of potential DNA methylation drivers (CpGs) for RELA (left panels) and PFA (right panels) recurrent tumors. **E** Schematic illustration of the data analysis steps to identify potential driver genes regulated by potential DNA methylation drivers. Differentially expressed genes (DEGs) (log2 fold change between tumor and normal tissues) were extracted from RNA-seq of the same set of tumors. DEGs discovered in patient tumors but absent in the matching PDOX tumors were filtered out to identity genes that contributed to PDOX tumorigenicity. **F** Correlation of expression and DNA methylation of potential genes were shown in the scatterplots for RELA (upper panel) and PFA (the lower panel). Those negatively correlated in RELA were highlighted in red (upper panel) and in PFA in blue (lower panel) and listed to the right of the plot.

tumors, have previously been associated with human cancers or ependymoma tumor dependency gene (*CACNA1H*)[11,65]

Despite the biological differences between RELA and PFA tumors, there were 7 downregulated genes shared by the two types of tumors (*ARRB1, KCNJ9, KIAAO513, KIF5C, SNAP25, SPTBN2,* and *TNR*) and they were all regulated by DNA hypermethylation (Supplementary Data 6 and 7). Worthy of note is that we also found over-expression of *WEE1*, of which multiple inhibitors have entered clinical trials[66,67], in 4/4 PFA and 3/4 RELA tumor sets (Supplementary Figs. 5C, 7B). Compared with RELA tumors, PFA tumors had nearly 3 folds more potential driver genes, most probably due to the differences of cell-of-origin. Altogether, this longitudinal DNA methylation analysis not only identified potential drivers (DMRs and genes) of relapses that were specific to or shared by RELA and PFA tumors, but also revealed the underlying mechanisms of the altered expressions in a set of relapse-related potential driver genes.

## Potential DNA methylation boosters of relapse were discovered in the serial relapses of ependymoma

Our collection of the multiple serially relapsing tumors from the same patients also presented an opportunity to examine DMCs that were newly acquired in the relapsed tumors. The DMCs that persisted in all the recurrences may have sustained (and boosted) tumor relapse and contributed to the increased tumorigenicity. To improve the level of confidence in discovering recurrent-specific DMCs, we included an additional four PFA and four RELA primary tumors from a previous study (GSE87779)[16] as a validation set (Fig. 4A, B, Supplementary Fig. 7D) and identified a set of RELA and PFA recurrent-specific DMCs in the relapse tumors in this public database (Supplementary Fig. 7D). Following our analysis of individual tumors, we extracted the DMCs that were shared by the 5 sets of RELA or the five sets of PFA recurrences (hereafter referred as potential DMC$^{Booster}$), and identified 296 HypoDMC$^{Booster}$ and 38 HyperDMC$^{Booster}$ in RELA; 165 HypoDMC$^{Booster}$ and 323 HyperDMC$^{Booster}$ in PFA recurrences (Fig. 4A). We applied the same strategy as detailed in Fig. 3 to identify the genes regulated by DMC$^{Boosters}$. In RELA tumors, we found 115 HyperDMRs$^{Booster}$ associated genes and 35 HypoDMRs$^{Booster}$ associated genes); whereas in PFA tumors 124 HyperDMRs$^{Booster}$ and 219 HypoDMRs$^{Booster}$ associated genes (Supplementary Data 9). Many of these genes could have been missed if only examine one recurrent tumor.

Since many of the relapsed tumors were not tumorigenic until late recurrences, we reasoned that some of the potential booster related genes contributed to the elevated tumorigenicity. Direct comparison between patient tumors and their PDOX tumors showed that only 21.7% (25/115) hyperDMR$^{Booster}$ and 28.6% (10/35) hypoDMR$^{Booster}$ related genes in RELA tumors were maintained in the PDOX tumors, whereas in PFA tumors, it was 89.5% (111/124) and 89.8% (188/219 genes) (Supplementary Data 9), which was significantly higher than that in RELA tumors ($P < 0.001$). These data suggested that the number of genes regulated by DMR$^{Boosters}$ was affected by the difference of cell-of-origin

and played a more important role in PFA recurrences, and PDOX tumor formation can be particularly helpful in identify tumorigenic DMR$^{Booster}$ related genes in RELA tumors.

To identify potential DMR booster driven genes with high stringency, we focused on the DEGs consistently present in all the relapsed tumors. Unlike the potential DMC-driver related genes, there were no shared potential booster genes between RELA and PFA tumors. In the RELA recurrent tumors, the expression of 10/10 (100%) hyperDMR-regulated genes (*NCDN, DES, MF12, CYB5A, BANK1, DLGAP4, SMTN, UCK1, WAC,* and *WWWC3*)[68] and 1/3 (33.3%) hypoDMR regulated genes (*PLEKHG1*) were negatively correlated with the corresponding DNA methylation changes (Fig. 4C, upper panel), accounting for 3.4% and 0.48% of the total down- and upregulated genes in RELA tumors. More importantly, all these genes were maintained in the PDOX tumors (Supplementary Data 9 and 10) which further supported their role in tumorigenicity. Functionally, some of the downregulated genes by hyperDMR have been associated with human cancers, including advanced stage of cancers (SMTN[69] and DES[70], BANK1[71]), autophage related death and prognosis of pancreatic cancer (CYB5A)[72,73] and aggressive phenotype of glioma and colorectal cancer recurrence (MFI2)[74,75]. As the only upregulated potential booster gene, *PLEKHG1* appeared to be an attractive therapeutic target. Although its biological function is not fully understood, a reverse correlation of *PLEKHG1* expression with poor survival in low grade gliomas has been noted (Supplementary Fig. 8A, B). In PFA tumors, we detected 41/66 (62.1%) hyperDMR down-regulated genes (including *HTR1A, GRM5, FGF5, GPR25* and *SHH*) and 58/89 (65.2%) hypoDMR upregulated genes (including *CAPS, ALDH3A1, FAM74A3, FOXJ1, EHF, ITGB5, NOTCH1, EPHA2,* and *SUFU*) (Fig. 4C, lower panel) (Supplementary Data 10), accounting for 3.2% and 4.5% of the total down- and upregulated genes in the PFA tumors. In addition to NOTCH[76], EPHA2[77] and SUFU[78] that are known to be involved in EPN biology, many of these genes (FOXJ1[79], ALDH3A1[80], EHF[81]) were associated with human cancers or brain tumors (Supplementary Fig. 7). Although the small number of dysregulated genes limited our capacity of detailed biological enrichment analysis, our discovery of their potentially roles in promoting EPN relapses is exciting and warrants future functional validation and drug development.

## Relapse predictors of DNA methylation can be identified from primary tumors at diagnosis

Since not all EPNs relapse, it is highly desirable to develop DNA methylation markers that can predict tumor recurrences when the tumor is diagnosed. We subtracted the primary tumor DMCs of EPNs that eventually recurred (i.e., primary-tumor$^{Eventually Recurred}$) from those in the primary tumors that did not relapse (primary tumor$^{Not-relapsed}$) over 10 years follow-up. Using UpsetR, we plotted the numbers of DMCs shared by primary-tumor$^{Eventually Recurred}$ or unique in primary tumor$^{Not-relapsed}$ (Fig. 5A, vertical histogram) as well as the DMCs from each primary tumor (Fig. 5A, horizontal histogram). We further hypothesized that the active predictors identified in the primary

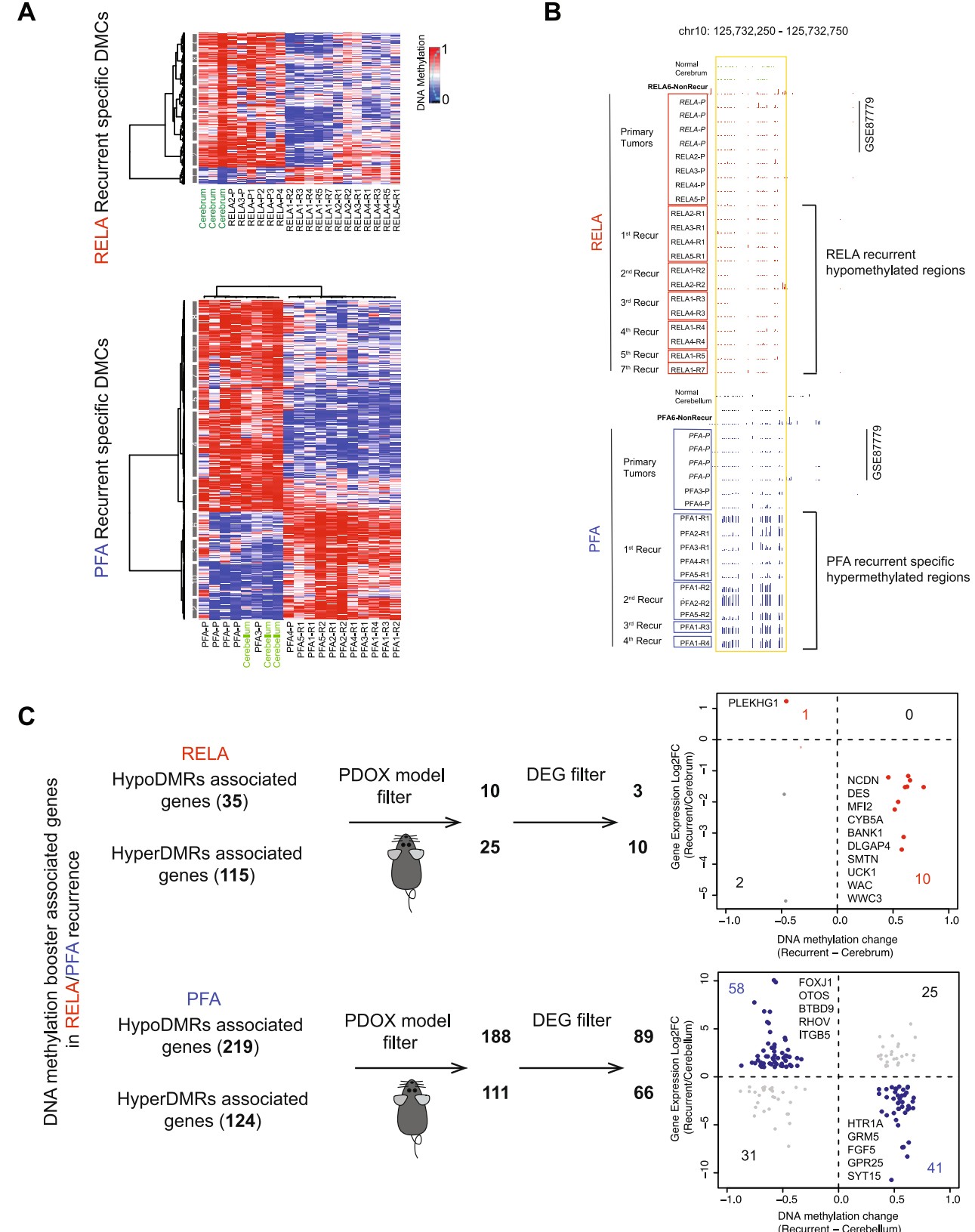

**Fig. 4 | Identification of potential DNA methylation booster of recurrent ependymoma. A** Heatmap showing the DMCs that were newly acquired in the recurrent (*R*) RELA (upper panel) and PFA (lower panel) tumors but absent in the primary (*P*) ependymomas. **B** Representative UCSC genome browser showing regions that was selectively hypermethylated in all recurrent tumors of a RELA recurrent tumors (*n* = 7) (upper panel) and PFA (*n* = 4) (*lower panel*) but not in their matching primary tumors. **C** Scheme showing the identification of recurrent-specific DMR (DNA methylation booster) associated genes. Hyper- and hypoDMR associated genes found in the patient tumors but not preserved in the matching PDOX models were filtered out. Differential expressions genes (DEGs) that were negatively correlated with DNA methylation were shown in the scatterplots with relative levels of change for RELA (red in upper panel) and PFA (blue in the lower panel) together with a list of top candidate genes.

tumors should also be present in the subsequent recurrent tumors. As shown in Fig. 5B, the frequency of the DMCs specific to the primary tumor[Eventually Recurred] were significantly higher than that in the primary tumors[Not-relapsed]. To increase the stringency of our data analysis, we focused on the DMCs that were selectively present in all (12 of 12) of the recurrent RELA tumors and all (10 of 10) recurrent PFA tumors (Fig. 5B, C), respectively; and narrowed the candidate predictor DMCs to 2207 hyper- and 791 hypoDMCs in RELA and 1305 hyper- and 2029 hypoDMCs in PFA tumors (Fig. 5D). From this list, we further ranked the DMCs (Fig. 5E) based on the differences of DNA methylation ratios between tumor and normal tissues and identified 7 DMCs in RELA and 22 DMCs in PFA tumors (with differences of DNA methylation ratio ≥0.8) as the top candidates of relapse predictor DMCs (Fig. 5E). Worthy of note is that some markers with the highest confidence, such as chr17-80943940-80943942 and chr5-92910051-92910053, exhibited wide-range of DNA methylation ratios, suggesting the possibility of sample variations. Despite the relatively small sample size, our discovery of this set of "recurrence-bound" Hyper- and HypoDMCs in the primary tumors is very encouraging. They provided proof-of-principle to support future extended studies on this important topic. New diagnostic predictors of recurrence can potentially cause a paradigm shift in the clinical care of childhood EPNs.

## Discussion

In this study, we improved the understanding of pediatric EPN relapse by using a collection of serially relapsing EPN tumors matched with primary tumor at diagnosis. Using a high-resolution analysis of DNA methylation together with RNA-seq, we showed that the molecular subtypes were maintained during long-term consecutive serial relapses and discovered that their epigenetic profiles progressively converged during serial relapses. Systematic analyses on animal tumorigenicity derived from the same panel of tumor samples revealed a significantly increased PDOX tumorigenicity in some late relapse tumors, which provided functional data supporting the progressive nature of EPN relapses. Parallel analysis of these PDOX models also bridged a gap between the epigenetic reprogramming and the increased tumorigenicity by fine-tuning the potential DMC drivers and boosters critical for EPN relapses.

Patient tumors represent the most reliable source for biological studies. One of the major challenges in understanding tumor relapse is the very limited availability of recurrent tumor tissues[78]. It requires a strong collaborative team and a long-term commitment. In the current study, we proactively collected and carefully followed 110 EPN patients over 13 years and successfully collected 10 (9%) sets of relapsed tumors. However, when compared with a relapse rate of 50% in childhood EPNs, we only captured <20% the recurrences. One major reason is that the treatment of EPN relapses does not include surgery or biopsy as the need of making diagnosis has already been met in the primary tumors and there is currently no sufficient justification for a routine "second look" of tumor pathology. Our discovery of epigenetic reprogramming in the recurrent tumors provided biological evidence to support biopsy or surgical resection of recurrent EPNs for updated molecular diagnosis and informed clinical treatment of recurrent EPNs.

The epigenetic convergence during repeated recurrences of pediatric EPNs indicates a decreased cellular heterogeneity of DNA methylation in the recurrent tumors, a result different from previous reports on the increased or expanded gene mutation loads in the recurrent tumors of other type of cancers[44]. One possible cause of this phenomenon is that clinical therapies, particularly radiation therapy, which remains the mainstay of clinical treatment in pediatric EPNs[1,2], selected or conditioned a subpopulation of surviving tumor cells for relapse. This result may be clinically important as it suggested a possibility of targeting a small set of epigenetic drivers of recurrence for significantly improved efficacy.

Tumor recurrence is propelled by a cascade of genetic and epigenetic events. Our identification of potential relapse drivers (the DMCs and genes that persisted from primary to all the relapsed tumors) and boosters (the DMCs and genes that only emerged in the relapsed tumors) deciphered a detailed long-range roadmap of pediatric EPN recurrences. However, childhood brains are often in different differentiation status, e.g., from 2–10 year old as in our cohorts. When combined with different time frame of recurrence, ranging from 1 to 13 years, some of the potential DMC drivers may be attributed to the patient specific-differentiation status of cerebrum and cerebellum. In addition to discovering a set of potential driver and booster genes for EPN relapses, one encouraging aspect of our finding is that many of the genes have already been involved in human cancer biology. For example, CACNA1H, a voltage-gated calcium channel, has been detected in breast cancer[56,82]; RARA, a retinoic acid receptor alpha, has exhibited important roles in leukemia and recently in medulloblastoma and glioma[83]; and HSPB8 (heat shock protein beta-8) promotes glioma growth and metastasis[61,84]. As the only potential booster gene in RELA relapses, PLEKHG1 (pleckstrin homology and RhoGEF domain containing G1) was significantly upregulated in gastric cancer plasma and associated with poor overall survival[85]. Identification of NOTCH, EPHA2 and SUFU as potential booster genes of PFA relapse not only suggested potential roles of these genes in recurrences, but also suggested a set of potential druggable targets for the difficult-to-treat PFA tumors[76,86].

Identification of markers that can predict tumor recurrence at the time of diagnosis is highly desirable. Similar to difficulties in obtaining tumor tissues from recurrent EPNs, it is also difficult to locate surgical samples from patients who remain tumor free for at least >5 years from diagnosis. Despite extensive efforts, we were only able to obtain 1 sample each of RELA and PFA tumors that did not recur 5–10 years. Our identification of DMC predictors of recurrence may help patient stratification for biologically based rational selection of treatment strategies. This set of proof-of-principle data should ignite broad interest in this field by analyzing larger collections of recurrent and non-recurrent samples to improve and validate the list of predictors. Despite sample variations of candidate predicators, it is highly desired that the number of predictors for EPN recurrence for each current or future molecular subtypes will decrease or be clinically applicable.

To translate biological studies from bench to clinic, we systematically implanted the same panel of tumor tissues into the matching locations in the mouse brains. Although it is time-consuming requiring nearly 13 years of continued efforts, we demonstrated the power of this strategy by discovering the elevated tumorigenicity of late recurrences, filtering out 6–10% the DMCs that were not directly involved in PDOX formation, and functionally validating potential DMC drivers that sustained the progression of recurrent EPNs. These set of data extended previous and our findings of chromosome 1q gain in promoting poor prognosis and potentially driving PFA tumorigenicity by providing a broader and higher resolution molecular signatures. This panel of clinically relevant animal models may facilitate the biological and preclinical studies of EPN relapses.

There are some limitations of our study. The number of patients is relatively small due to the rarity of pediatric EPNs, and we hope our finding will facilitate the biopsy or surgery on recurrent tumors to better understand tumor biology and increase the availability of recurrent tumors. Tumorigenicity can be affected by multiple factors. Although we have tried to standardize the protocol, there might be other determinants of PDOX tumor formation in addition to the genetic/epigenetic changes that were addressed in our study. Analysis of single cells may shed light on the cellular heterogeneity and driver cells of EPN recurrences. Emerging data have suggested potential roles of non-CpG methylation in brain development and cancer biology[40,41,43,87]. In addition, more cases and higher resolutions (e.g., whole-genome methylation sequencing) are needed to support

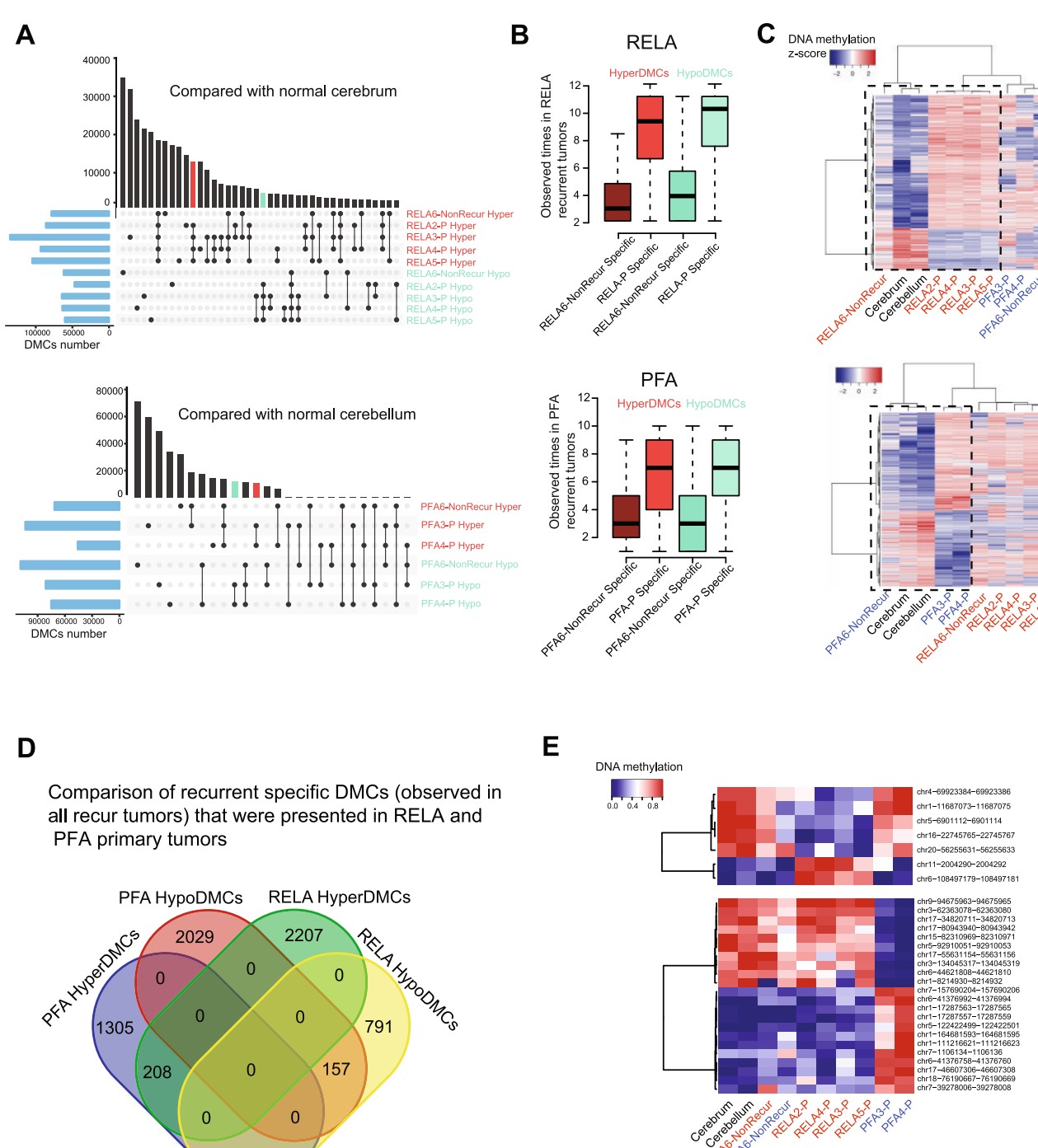

**Fig. 5 | Identification of DNA methylation predictors of recurrence in primary ependymoma tumors. A** UpSet R plot showing the hyper- and hypo-methylated CpGs sites (DMCs) in RELA (upper panel) and PFA (lower panel) primary tumors (-*P*) as compared with normal cerebrum or cerebellum tissues. The horizontal histogram represents the number of DMCs in each comparison between primary tumor and normal brain tissues; the vertical histogram represents the number of DMCs shared by tumors marked by connected dots. Red and green bar highlights the hyper- and hypoDMCs that were shared by all the tumors that eventually recurred (primary-tumor^Eventually recurred) but not in the normal brain tissues and the non-recurrent reference tumors. **B** Bar graph showing the frequencies of hyper- and hypoDMCs specific to ependymoma primary tumors that eventually recurred (-*P-Specific*) and that did not relapse (-*NonRecur Specific*) in the RELA recurrent ($n = 12$) and PFA ($n = 10$) recurrent tumors. Boxplots indicate median, first and third quartiles (Q1 and Q3), whiskers extend to the furthest values; the uppermost and lowest line indicates the maximum and minimum values, respectively. In RELA tumors, the

numbers of HyperDMCs analyzed are RELA6- NonRecurSpecific ($n = 10{,}309$ CpGs), RELA-P-specific ($n = 357{,}141$ CpGs); whereas of the HypoDMCs are RELA6-NonRecurSpecific ($n = 17{,}188$ CpGs) and RELA-P-specific ($n = 507{,}987$ CpGs). In PFA tumors, the HyperDMCs analyzed are PFA6-NonRecurSpecific ($n = 24{,}296$ CpGs) and PFA-P-specific ($n = 349{,}746$ CpGs); and HypoDMCs PFA6NonRecurSpecific ($n = 40{,}525$ CpGs); PFA-P-specific ($n = 451{,}426$ CpGs). **C** Heatmap showing the DNA methylation ratios of DMCs specific to RELA (upper panel) or PFA (lower panel) primary tumors^Eventually recurred exist in all recurrent tumors. Black dash box shows that these DMCs have similar DNA methylation ratios among RELA-/PFA-NonRecur primary tumor, normal cerebellum and cerebrum, but different from the primary tumors Eventually recurred. **D** Venn diagram representing the number of over-lapped and specific hyper/hypoDMCs (from **C**) between RELA and PFA. **E** Heatmap showing the top CpGs' DNA methylation ratios with highest confidence that can potentially predict recurrence from primary tumors of RELA (upper panel) and PFA (lower panel) ependymoma.

our finding regarding the decreased mCpA in PFA and RELA EPN (regardless of relapses or not).

In summary, we defined a roadmap of the epigenetic progression in patient-matched serial EPN recurrences, discovered convergent DNA methylation profile as a feature of serial relapses, and suggested a set of potential DNA methylation drivers and boosters that sustained and promoted recurrence. We also uncovered an increased tumorigenicity in late recurrences that paralleled the epigenetic progression, established a panel of PDOX models and demonstrated the power of these models in fine-tuning the list of potential drivers and boosters of EPN relapses. Our findings may improve the insight about the underlying epigenetic mechanisms of EPN recurrences.

## Methods

### Tumor tissues from childhood EPN patients

Signed informed consent was obtained from the patient or their legal guardian prior to sample acquisition following an Institutional Review Board (IRB) of Baylor College of Medicine approved protocol (H-4844). Thirty-three freshly resected EPN tumor specimens from 10 patients undergoing serial surgical resections of primary and recurrent tumors (3.67 ± 1.76/per patient) over a period of 13 years (5.8 ± 3.8) at Texas Children's Hospital were obtained for this study. Four samples of normal brain tissues (two cerebral and two cerebellar tissues) obtained from warm (<6 h) autopsy of children were included as the normal references. The patients' demographic and clinical information is described in Table 1. All the samples were subjected to pathological diagnosis and graded following the WHO system. Tumor tissues were divided into two portions for processing. One part was snap frozen in liquid nitrogen and preserved in the −80 °C freezer. The 2nd part of fresh tumor tissues was washed and minced with fine scissors into small fragments. Single cells and small clumps (3–5 cells per clump) of tumor cells were collected with a 35 μ cell strainer, resuspended in DMEM growth medium to achieve a final concentration of $1 \times 10^8$ live cells per ml, as assessed by trypan blue staining, and transferred to animal facility on ice.

### Global reduced representation bisulfite sequencing (RRBS)

Genomic DNA was extracted with Allprep DNA/RNA mini kit (Qiagen). and concentrations measured using the Qubit® dsDNA BR Assay Kit (Thermo Fisher Scientific), followed by DNA quality assessment with the Fragment Analyzer™ and either the DNF-487 Standard Sensitivity or the DNF-488 High sensitivity genomic DNA Analysis Kit (Advanced Analytical). RRBS libraries were prepared using the Premium Reduced Representation Bisulfite Sequencing (RRBS) Kit (Diagenode Cat# C02030033), according to the manufacturer's protocol. 100 ng of genomic DNA were used to start library preparation for each sample. Following library preparation, samples were pooled together either by 8 (mouse contamination up to 10%), 7 (mouse contamination up to 25%), 5 (mouse contamination up to 40%), or 4 (mouse contamination up to 55%). In total, 16 pools were prepared. PCR clean-up after the final library amplification was performed using a 1.5× beads:sample ratio of Agencourt® AMPure® XP (Beckman Coulter). RRBS library pools quality control was performed by measuring DNA concentration of the pools using the Qubit® dsDNA HS Assay Kit (Thermo Fisher Scientific), and the profile of the pools was checked using the High Sensitivity DNA chip for 2100 Bioanalyzer (Agilent). Each RRBS library pool was deep sequenced on one lane HiSeq3000 (Illumina) using 50 bp single-read sequencing (SR50).

### RRBS data generation and processing

Raw RRBS FASTQ files were mapped to NCBI Human Reference Genome Build GRCh37 (hg19) using BSMAP (v2.9) RRBS mode[20]. DNA methylation ratio and differential methylated cytosine (DMCs/DMRs) were analyzed by using MOABS (v1.2.9)[21]. CpG sites with five or more reads covered were used for downstream analysis. Bisulfite conversion

rates were estimated on the basis of lambda phage genome spike-ins. The bedGraph files including single base pair DNA methylation ratios were transformed to bigwig file format which can be visualized using the UCSC genome browser. DNA methylation heatmaps were plotted using R package https://www.rdocumentation.org/packages/heatmap3/versions/1.1.6/topics/heatmap3 heatmap3 by taking the shared CpGs among all the samples as input. DNA methylation phylogenetics analysis was performed by using R package ape[22]. To compare multiple groups' DMCs, we merge all the DMCs in all two-group comparisons (union DMC sets). The UpSetR[23] package was used to visualize the union of DMC sets among multiple two-group comparisons. To analyze dynamic changes of DMCs among tumor recurrent, we first separate DMCs to three categories (Hyper; Hypo; NoChange) based on adjacent two recurrent stages. Then, by considering four adjacent recurrent stages (P vs. cerebellum; R1 vs. cerebellum; R2 vs. cerebellum and R3 vs. cerebellum), we filtered out those DMCs with hyper/hypo and hypo/hyper switch between any two adjacent stages due to the small numbers. We finally separated DMCs into seven categories: consistent Hyper; consistent Hypo; Gain Hyper; Gain Hypo; Loss Hyper; Loss Hypo and switch (between hyper/hypo and NoChange). R alluvial package (https://www.rdocumentation.org/packages/alluvial/versions/0.1-2/topics/alluvial) was used to visualize the dynamic changes of DMCs along tumor recurrence. Colored density scatterplot of DNA methylation ratios was performed by using R package smoothScatter (https://www.rdocumentation.org/packages/graphics/versions/3.6.1/topics/smoothScatter). GREAT[24] was used to predict DMRs' functions. The analysis codes are available at https://github.com/lijiacd985/Mmint.

To infer DNA copy number status, particularly chromosome 1q, we applied CNVkit (https://cnvkit.readthedocs.io/en/stable) to infer CNV using RRBS data for PFA human and PDX samples. The CNVkits calculate normalized coverage in bin-level then it removes the systemic bias (such as CG content) use circular binary segmentation (CBS) to infer discrete copy number regions as segments.

### RNA-seq analysis

RNA-seq libraries for transcriptome analysis were prepared using the TruSeq RNA Sample Preparation Kit (Illumina) and Agilent Automation NGS system per manufacturers' instructions. Sample prep began with 1 μg of total RNA from each sample. Poly-A RNA was purified from the sample with oligo dT magnetic beads, and the poly(A) RNA was fragmented with divalent cations. Fragmented poly-A RNA was converted into cDNA through reverse transcription and were repaired using T4 DNA polymerase, Klenow polymerase, and T4 polynucleotide kinase. 3' A-tailing with exo-minus Klenow polymerase was followed by ligation of Illumina paired-end oligo adapters to the cDNA fragment. Ligated DNA was PCR amplified for 15 cycles and purified using AMPure XP beads. After purification of the PCR products with AMPure XP beads, the quality and quantity of the resulting.

FastQC (http://www.bioinformatics.babraham.ac.uk/projects/fastqc/) was used to do quality checks for raw fastq files. Raw FASTQ files were aligned to NCBI Human Reference Genome Build GRCh37 (hg19) using HISAT2[16] with default settings. The uniquely mapped reads were used for downstream analysis. HTSeq[17] was used to count the reads count mapped in exon regions for each gene. Read counts matrix (row as genes; column as samples) were inputted to DESeq2[18] to identify differentially expressed genes (DEGs). We consider genes with ≤FDR ≤ 0.05 and fold change ≥ 2 folds as DEGs. Principal component analysis of DEGs was performed using R package DESeq2. DEGs' function enrichment was using GSEA[19].

### Development of patient-derived orthotopic xenograft (PDOX) mouse models

The SCID mice, NOD.129S7(B6)-*Ragl^{tm1Mom}*/J (Jax Laboratory), mice were bred and housed in a specific pathogen-free (SPF) animal

facility at Texas Children's Hospital in Houston or Lurie Children's Hospital in Chicago. All the experiments were conducted using Institutional Animal Care and Use Committee (IACUC) approved protocols from Baylor College of Medicine or Northwestern University. The Office of Laboratory Animal Welfare in both institutions are fully accredited from the Association for Assessment and Accreditation of Laboratory Animal Care International (AAALACI). The vivarium water, temperature, and light cycles are controlled by centralized computers. The vivarium is staffed by full time veterinarians and support personnel that administer a complete program of veterinary care. Tumor tissues from primary and surgical transplantation of tumor cells into the mouse brain was performed using a free hand implantation strategy[29–31]. Both male and female mice, aged 5–8 weeks (to simulate the developing brain in children), were anesthetized with inhaled isoflurane and/or pentobarbital (50 mg/kg, *i.p.* injections). Each animal will be given a pain medication at time of surgery before the intracranial injection of tumor cells. Tumor cells ($1 \times 10^5$) were suspended in $2\,\mu l$ of culture medium and injected into the right frontal/temporal region (1 mm to the right of the midline, 2.5 mm anterior to the lambdoidal suture) or right cerebellar lobe ((1 mm to the right of the midline, 1 mm posterior to the lambdoidal suture) and 3 mm deep via a $10\,\mu l$ 26-gauge Hamilton Gastight 1701 syringe needle[29–31]. After tumor injections, animals will be monitored daily for 3–4 days. After the initial period of daily observation, all tumor-bearing animals will be monitored for symptoms of discomfort or pain. A veterinary intervention will be elicited if an animal is unable to eat or move or showing behavior such as huddled posture or self-mutilation, or any signs of infection (such as eye or ear). If an animal develops torticollis, uncontrolled circling or other signs of neurologic deficits such as limb paralysis, or loss of body weight (>15%), or become moribund, the animal will be euthanized through intraperitoneal injection of Euthasol (pentobarbital sodium and phenytoin sodium) at 150 mg/Kg to introduce deep anesthesia before the whole mouse brain is removed for histopathologic examination. Those mice without any neurological deficit after 12 months were euthanized and examined for tumor development. To perform serial subtransplantations, whole brains of donor mice were aseptically removed, coronally cut into halves, and transferred back to the tissue culture laboratory. Tumors were then dissected under the microscope, mechanically dissociated into cell suspensions, counted and injected into the brains of recipient SCID mice as described above[30].

### Fluorescence in situ hybridization (FISH) analysis

FISH analysis was performed on $5\,\mu m$ paraffin embedded sections slides[48] using Vysis/Abbott Molecular (Des Plaines, IL) dual color probes targeting chromosome 1p36.3/TP73 and 1q25.2/ANGPTL loci, with the 1p36.3/TP73 locus labeled with spectrum red and the 1q25.2/ANGPTL labeled with spectrum green, for the detection of copy number alterations of both loci, following standard laboratory procedures in the clinical cytogenetics laboratory. A total of 100 non-overlapping cells were evaluated by two technologists independently. The average signals for both 1p36.3/TP73 and 1q25.2/ANGPTL were calculated. The signal ratio of 1q/1p ≥ 2.0, is interpreted as gain or amplification of 1q25.2/ANGPTL.

### Reporting summary

Further information on research design is available in the Nature Portfolio Reporting Summary linked to this article.

## Data availability

The raw RNA-seq and methylation data generated by this study are available from the NCBI under accession number GSE156619. The RELA and PFA signature gene list is from public dataset GSE64415)[16,39,88]. Additional RELA and PFA primary tumor WGBS data are from public dataset GSE87779[10] and public DNA methylation array data for RELA and PFA relapse samples is from GSE65362[16]. Source data are provided as a Source Data file. The remaining data are available within the Article, Supplementary Information or Source Data file. Source data are provided with this paper.

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

## Acknowledgements

This work was funded by NIH RO1 CA185402 (X.N.L.), St. Baldrick's Foundation (Grant 2532341503, J.M.S.), Golfers against Cancer (X.N.L.), Childhood brain tumor foundation (X.N.L.), National Brain Tumor Foundation (X.N.L.), The Science development program of Guangzhou 201707020001(Y.F.X.), CPRIT core grant RP150578 and RP200668 (C.S. and P.A.D.).

## Author contributions

Concept and experimental design (S.Z., Y.F.X., and X.L.N.), data collection (S.Z., H.Z., L.Q., Y.D., M.K., F.D.B., S.X., Y.H., J.L., H.L., P.B., J.M.F., M.L., P.G., A.C.V., S.S., G.B., S.R.D., L.D., C.S., J.Y., P.J.D., X.L., M.C., D.W.P., and L.P.), histopathology evaluation (A.A.), data analysis (J.L., D.S., T.M., S.Z., W.Y.T., Y.H., and X.N.L.), manuscript preparation (S.Z., J.L., W.Y.T., and X.N.L.).

## Competing interests

M.L., P.G., A.-C.V., S.S., and G.B. were employees of Epigenetic Services, Diagenode, Liège, Belgium. The remaining authors declare no competing interests.

## Additional information

[1]Pre-clinical Neuro-oncology Research Program, Texas Children's Hospital, Baylor College of Medicine, Houston, TX 77030, USA. [2]Texas Children's Cancer Center, Texas Children's Hospital, Baylor College of Medicine, Houston, TX 77030, USA. [3]Jane and John Justin Neurosciences Center, Cook Children's Medical Center, Fort Worth, TX 76104, USA. [4]Hematology and Oncology Center, Cook Children's Medical Center, Fort Worth, TX 76104, USA. [5]Center for Epigenetics & Disease Prevention, Texas A&M University, Houston, TX 77030, USA. [6]Center for Translational Cancer Research, Institute of Biosciences and Technology, Texas A&M University, Houston, TX 77030, USA. [7]State Key Laboratory of Respiratory Disease, National Clinical Research Center for Respiratory Disease, Guangzhou Institute of Respiratory Health, the First Affiliated Hospital of Guangzhou Medical University; and Guangzhou Laboratory, Bioland, 510120 Guangzhou, Guangdong, P. R. China. [8]Program of Precision Medicine PDOX Modeling of Pediatric Tumors, Division of Hematology-Oncology, Neuro-Oncology & Stem Cell transplantation, Ann & Robert H. Lurie Children's Hospital of Chicago; Department of Pediatrics, Northwestern University Feinberg School of Medicine, Chicago, IL 60611, USA. [9]Department of Neurosurgery and Brain and Nerve Research Laboratory, the First Affiliated Hospital, and Department of Neurosurgery, Dushu Lake Hospital, Suzhou Medical College, Soochow University, 215007 Suzhou, P. R. China. [10]Humphrey Oei Institute of Cancer Research, National Cancer Center Singapore, Singapore 169610, Singapore. [11]Cancer and Stem Cell Biology Program, Duke-NUS Medical School Singapore, Singapore, Singapore. [12]KK Women's & Children's Hospital Singapore, Singapore, Singapore. [13]Institute of Molecular and Cell Biology, A*STAR, Singapore, Singapore. [14]Department of Pathology, Texas Children's Hospital, Baylor College of Medicine, Houston, TX 77030, USA. [15]Epigenetic Services, Diagenode, Liège, Belgium. [16]State Key Laboratory of Oncology in South China, Collaborative Innovation Center for Cancer Medicine; Department of Radiation, Sun Yat-sen University Cancer Center, 510060 Guangzhou, Guangdong, P. R. China. [17]Clinical Cytogenetic Laboratory, Department of Pathology, Northwestern University Feinberg School of Medicine, Chicago, IL 60611, USA. [18]These authors contributed equally: Sibo Zhao, Jia Li, Huiyuan Zhang. ✉e-mail: deqiangs@zju.edu.cn; xiaonan.li@northwestern.edu

