## [Peer Review File · Nature Communications]

Epigenetic Alterations of Repeated Relapses in Patient-matched Childhood EpendymomasREVIEWER COMMENTS

Reviewer #1, expertise in ependymomas, genomics, mouse models of brain tumours (Remarks to the Author):

In the present manuscript, Zhao et al. performed single-base resolution DNA methylation analysis and report convergency of DNA methylation profiles during the progression of tumor recurrences. They identified differentially expressed genes which are regulated by driver DMRs and booster DMRs. Moreover, they also identified potential relapse predictors in RELA and PFA tumors. In general, this manuscript provided a new insight into epigenetic regulation of repeated recurrences in ependymoma. In general, the data itself for study mechanisms underlie recurrence in EPN is valuable. However, the findings are preliminary and most of the statements in this study are derived from bioinformatic results, no strong experimental evidence is provided. Several points that need to be addressed are listed below and hope those comments would be helpful.

Major concerns:

1. The authors determined two subtypes of EPNs (PF and RELA) using the Phylogeny tree and PCA analysis of the methylation and RNA-seq dataset, the results indicated that these two subsets are indeed distinctive. However, there is no information of the brain regions where the samples were collected, the authors should ensure the heterogeneity are not from the brain region differences, the compassion is only reasonable when the samples are from the same/similar brain regions.
2. In Fig1A, the author reported "Among the 5 patients whose primary tumors (n=1 patient: PFA3) or early recurrent tumors (n=4 patients: RELA1, PFA5, PFA2, PFA1) did not form xenografts, their late recurrent tumor(s) formed PDOX tumors". It would be more convincing to provide experimental images besides the schematic diagram shown in Fig1A.
3. Fig.1E-F, it is not very clear how the genes were selected to show in the heatmaps. Are they representative enough to convey the message claimed in the main text?
4. Please elaborate on the saying "This finding is new", there is no enough context for readers to buy it.
5. In Figure2a, how the Pearson correlation was calculated if the adjacent previous relapsing tumor was missed? Especially for instance the RELA1, the primary and 1st relapsing samples are missed, was the first Pearson correlation calculated between the 2nd relapsing sample and the control?
6. For the alluvia plots, the authors should give more details how the stream number were used for consistent/gain and loss Hypo/Hyper. Why the GO enrichments of GSEA analysis of shared DRMS associated genes in RELA and PFA are not related to tumor or diseases, only associated with neural system development? Are the tumor signature genes from Figure1EF are in the DEG lists?
7. In Fig3, the author claimed "the strong negative correlations between the DMRs and their target genes indicated a causative role of the DMR drivers in regulating the gene expressions". This statement is overstated and not supported by experimental evidence.
8. The definition of driver and booster is not solid enough to me, the conclusion inferred from them is therefore suspicious. Can the authors provide some reference or a third-party method to prove that the DMCs "that persisted in all the recurrences may have sustained (and boosted) tumor relapse and contributed to the increased tumorigenicity" indeed have a booster role?
9. In Figure4C, could the author also show the biological enrichments of the DEGS that were negatively correlated with DNA methylation for RELA and PFA. What are the deep implications of these genes?
10. In Fig5, the author reported several potential relapse predictors for in RELA and PFA tumors. As mentioned in the text, the sample size is relatively small. Then, how would the number of identified relapse predictors change as sample size changes? Would the number of candidates DMCs decrease as the sample size increases? If it's the case, how reliable the predictors are, since there could be large sample variation.

Minor concerns:

- 1. In Figure1c, it makes more sense to show the column dendrogram as well. In Figure1d, there are two RELA samples are closed to the Cerebrum samples, could the authors show the names of the two samples, and give an explanation of this situation? In Figure1EF, how the authors determine the signature genes?**
- 2. In figure1a, how to explain the correlation value decrease for the RELA2 between the 1st and 2nd relapsing?**
- 3. Scale bar in Figu2B is missing.**
- 4. In Figure2D, could the authors show the intersectional correlations between patient tumor and PDOX models? Such as the R7 RELA1 patient tumor to the R1 RELA4 PDOX model, et al.**
- 5. could the authors give explain the deep implication of the number of DEG differences in the upset plots?**
- 6. Scale bar in Fig4A is missing.**

Reviewer #2, expertise in ependymomas, genomics, DNA methylation (Remarks to the Author):

The study "Epigenetic Drivers and Boosters of Repeated Relapses in Patient-matched Childhood Ependymomas" by Zhao, Li, Zhang et al., presents an invaluable, rare, and well curated cohort of 30 ependymoma samples including patient-matched repeated relapses over 13 years. Similarly, the described cohort of 13 new ependymoma PDX models is of great value for future functional validation of new targets and treatments.

The authors rightfully focus on the question of why many ependymoma tumors recur. To uncover new molecular mechanisms of tumor recurrence, the authors decided to closely examine genome-wide DNA methylation patterns and generated a rich data set of gene expression and reduced representation bisulfite sequencing (RRBS). In line with previous reports, DNA methylation remains very stable throughout disease progression. It is really fascinating how stable the DNA methylation profiles are and even converge during repeated relapses. At the same time, this property somewhat limits the space for novel insights into ependymoma tumor biology by focusing on DNA methylation.

The authors have extensively used many different standard methods to analyze DNA methylation, such as the analysis of DMCs, DMRs, non-CpG methylation, and the association of DNA methylation patterns near transcription start sites with gene expression changes. The results are a detailed encyclopedic review of DNA methylation patterns in primary and relapse PFA and RELA ependymoma tumors.

Despite the limited new insights into the molecular mechanisms that promote tumor recurrence and the lack of experimental validations, the study has a high impact as a resource due to the invaluable sample cohort and the epigenetic data obtained. Please find detailed comments below.

1) "Driver" methylation events are defined, but there is no functional evidence of these regions' tumorigenicity. PDX models are presented as validation, but these models only show that the tumor is tumorigenic, not that DNA methylation causes that tumorigenicity. Functional validations are likely beyond the scope of the study, which is

ok, but DNA methylation needs to be consistently described as a possible driver rather than a driver.

2) I understand that abnormal DNA methylation present in primary tumors is defined by a comparison to the normal cerebellum/ cerebra tissue. This needs to be described more clearly in the main text.

3) A statistical problem is the treatment of dependent samples as independent. Eg., the authors claim to have found a set of methylation markers for RELA EPN relapse, but this set is dominated by markers specific to a single patient's multiple relapses. Given the rarity of EPN and their small sample size, perhaps this is unavoidable. However, these markers need to be tested on an independent dataset of EPN tumors.

4) Page 6 "This finding is new and they demonstrated the maintenance of EPN molecular subtypes during repeated relapses (≥ 2) despite years of chemo- and/or radiation therapies (Fig 1A)." This is very impressive especially for the serial analysis. However, it has previously been shown that DNA methylation remains highly similar in the relapse compared to primary ependymomas.

5) Page 6 "For non-CpG methylation, the primary and recurrent tumors were found to have a dramatically decreased mCpA levels in both RELA and PFA tumors (Fig. S1F)." There is a striking difference of non-CpG methylation in the tumors as compared to the normal cerebellum/ cerebrum. Can the authors speculate what that means? Are these the same regions defined as abnormal DMCs/ DMRs?

6) It is very impressive to see that late recurrent tumor(s) are much more likely to develop as PDX tumors. However, it's not evident that "convergent epigenetic reprogramming" promotes "tumorigenicity in EPN relapses". Gain of chromosome 1q has been identified as a genetic marker associated with recurrence and poor survival in PFA ependymoma tumors. For PFA ependymoma, the emergences of 1q gains might drive tumorigenicity instead of DNA methylation. Have the authors attempted to identify chr1q copy number status using the RRBS data and are there more frequent chr1q gains in the PFA relapses that grow out as PDX tumors?

7) "These DMCs persisted from the primary tumors to late relapses, thereby constituting novel DNA methylation driver signatures of EPN relapse." What we know is that these CpGs are consistently more/ less methylated in EPN compared to normal tissue but that doesn't mean those drive EPN relapse. It could be that whole cerebrum and whole cerebellum are in a different differentiation status than the actual EPN precursor cells and there might still be something completely different that drives tumor progression.

8) Fig.3 F: x- and y-axes need a separate legend for RELA (Primary or recurrent – cerebellum/ cerebra)

9) "Most of the DMR regulated genes were newly discovered for EPN relapses. Their potential in as relapse driver genes was further enhanced by the fact that many of them, including CACNA1H, SLC12A7, CSPG4, RARA in RELA, and HSPB8, ITGB4, FAT1 in PFA tumors, have previously been associated with human cancers." Many of these gene have previously been described in ependymoma, such as CACNA1H and WEE1, and some of them were experimentally validated as ependymoma tumor dependency genes.

10) "longitudinal analysis of consecutive, serially relapsing patient tumor samples will enable the separation of epigenetic driver(s) from transient random alterations". Just because a mutation is conserved in subsequent relapses does not make it pathogenic; this would imply that non-pathogenic passenger mutations are somehow lost in subsequent relapses, which is not true.

REVIEWER COMMENTS

Reviewer #1, expertise in ependymomas, genomics, mouse models of brain tumours (Remarks to the Author):

In the present manuscript, Zhao et al. performed single-base resolution DNA methylation analysis and report convergency of DNA methylation profiles during the progression of tumor recurrences. They identified differentially expressed genes which are regulated by driver DMRs and booster DMRs. Moreover, they also identified potential relapse predictors in RELA and PFA tumors. In general, this manuscript provided a new insight into epigenetic regulation of repeated recurrences in ependymoma. In general, the data itself for study mechanisms underlie recurrence in EPN is valuable. However, the findings are preliminary and most of the statements in this study are derived from bioinformatic results, no strong experimental evidence is provided. Several points that need to be addressed are listed below and hope those comments would be helpful.

Response: We appreciate the insightful comments from the reviewers. They really are very helpful and surely make our manuscript stronger.

Major concerns:

1. The authors determined two subtypes of EPNs (PF and RELA) using the Phylogeny tree and PCA analysis of the methylation and RNA-seq dataset, the results indicated that these two subsets are indeed distinctive. However, there is no information of the brain regions where the samples were collected, the authors should ensure the heterogeneity are not from the brain region differences, the compassion is only reasonable when the samples are from the same/similar brain regions.

Response: We agree with the reviewer that the location of ependymoma is very important. We originally summarized the clinical information in Supplemental Table 1 (which was clearly not the best way of data presentation). Additionally, we brought Table 1 into the main text and updated it by including the 4 normal brain tissues obtain from autopsy. The updated **Table 1** is attached. And, all the PFA tumors were located in the posterior fossa (cerebella) and RELA in cerebral hemisphere.

Table 1: Summary of clinical information of the ependymoma patients and the autopsied normal tissues.

Recurrent Tumors							
Patient ID (Dx)	Age (years)	Gender	Tumor Location (primary & recurrent)	Total Recurrence Number	Time to First Recurrence	Time to Last Recur	Patient Status
PFA1	3	Female	Posterior fossa	3	14.7 months	35.8 months	Alive
PFA2	7	Male	Posterior fossa	2	56.9 months	90.3 months	LTF
PFA3	2	Male	Posterior fossa	1	43.6 months	N/A	Alive
PFA4	4	Male	Posterior fossa	2	13 months	33.2 months	Alive
PFA5	8	Female	Posterior fossa	3	60.9 months	131.5 months	LTF
RELA1	8	Female	Right fronto- parietal	7	60 months	170.4 months	Deceased
RELA2	10	Male	Right frontal	2	22.2 months	32.2 months	LTF
RELA3	7	Male	Right frontal	1	62.7 months	N/A	Deceased
RELA4	2	Male	Left frontal	4	12.4 months	44.3 months	Deceased
RELA5	6	Male	Right parietal	1	45 months	N/A	Deceased
Non-recurrent Tumors							
Patient ID	Age (years)	Gender	Tumor Location	Total Recur Number	Total Follow Up Duration		Patient Status
PFA6	2	Male	Posterior fossa	0	131 months		Alive
RELA6	6	Male	Right frontal	0	89 months		LTF
Normal Brain Tissues							
Tissue ID	Age (years)	Gender	Tissue Location	Source			
A1429-NC	9	Male	Right frontal	Autopsied			
A1429-NCb	9	Male	Right cerebellar	Autopsied			
A1123-NC	5	Male	Right frontal-parietal	Autopsied			
A1123-NCb	5	Male	Right cerebellar	Autopsied			

Note: Dx: diagnosis; Recur: recurrence; NC: normal cerebrum; NCb: Normal cerebellum; LTF: Lost to follow-up.

2. In Fig1A, the author reported "Among the 5 patients whose primary tumors (n=1 patient: PFA3) or early recurrent tumors (n=4 patients: RELA1, PFA5, PFA2, PFA1) did not form xenografts, their late recurrent tumor(s) formed PDOX tumors". It would be more convincing to provide experimental images besides the schematic diagram shown in Fig1A.

Response: Yes, we agree with the reviewer that this is a very good approach. We have thus relocated the tumor histology (original Fig 2C) to Figure 1B, and added five sets of H&E stained whole mouse brain sections together with three enlarged (10x) images to show the location of intra-cerebellar (ICb) and intra-cerebral (IC) xenograft tumors of early (small) and late (big) growth as well as a representative image showing CSF spread (please see the inserted images below for Fig 1A and 1B). Our plan is to report detailed cellular and molecular characterizations of these models in a separate paper in the near future.

3. Fig.1E-F, it is not very clear how the genes were selected to show in the heatmaps. Are they representative enough to convey the message claimed in the main text?

Response: In Figure 1E-F, these signature genes were selected from a previously published database GSE64415, which is primarily composed of primary tumors of ependymoma. The genes themselves were identified/defined by the authors of this database, and the levels of expression were derived from our dataset. Our goal is to make use of the independent set of data to validate our results.

4. Please elaborate on the saying "This finding is new", there is not enough context for readers to buy it.

Response: We believe the reviewer was referring to this sentence in page 6 "This finding is new and they demonstrated the maintenance of EPN molecular subtypes during repeated relapses (≥ 2) despite years of chemo- and/or radiation therapies (Fig 1A)." To avoid ambiguity, we revised it to the following on Page 6:

These set of data demonstrated that the maintenance of EPN molecular subtypes during repeated relapses (≥ 2) during years of chemo- and/or radiation therapies. (Fig 1A)

5. In Figure2a, how the Pearson correlation was calculated if the adjacent previous relapsing tumor was missed? Especially for instance the RELA1, the primary and 1st relapsing samples are missed, was the first Pearson correlation calculated between the 2nd relapsing sample and the control?

Response: Thanks for pointing this out. In the event an adjacent previous relapsing tumor was missing, we used available previous samples to calculate the correlation. And yes, it is correct in RELA1. We calculated first Pearson correlation between the 2nd relapsing sample and the control since the primary and 1st relapsing samples were missed. The purpose of such analysis is to identify the longitudinal dynamic changes of DNA methylation along tumor recurrence. We added the following sentence to clarify our approach on Page 6:

In the event an adjacent previous relapsing tumor was missing, a previously available sample was used to calculate the correlation.

6. For the alluvia plots, the authors should give more details how the stream number were used for consistent/gain and loss Hypo/Hyper. Why the GO enrichments of GSEA analysis of shared DRMS associated genes in RELA and PFA are not related to tumor or diseases, only associated with neural system development? Are the tumor signature genes from Figure1EF are in the DEG lists?

Response: 1) For the alluvia plots, we first compare every primary and recurrent tumor sample with the normal tissue control to identify DMCs, followed by comparing these DMCs between adjacent primary and recurrent tumor samples to identify the shared DMCs between any two adjacent samples and the specific DMCs for certain samples. The CpGs that were maintained with increased/decreased DNA methylation levels in all the recurrent samples (compared to control) were defined as the consistent gain or loss of Hyper/Hypo methylations. **2)** For the GSEA analysis, we used BP (Biological Process) database, which is predominately enriched with adult cancers with limited numbers of pediatric brain cancers. The biological differences between adult and pediatric cancer may have contributed to our analyzed results. Hopefully, the near future completion of pediatric cancer data commons and other shared database should shed more lights on pediatric cancers. **3)** As for the question “Are the tumor signature genes from Figure1EF are in the DEG lists?”, we found that the overall percentage is low. By comparing the 42 PFA/ 42 RELA signature genes (derived from Fig 1E, F) with consistent DEGs during PFA/RELA relapse, we found that **a)** in PFA tumors, there were three genes (IGF2BP3, LAMA2 and PDE3B) overlapped with the consistently decreased DEGs (n=1290) and eight genes (GORASP1, IGSF1, MECOM, MST4, PLIN5, PPARG, TKTL1 and VIPR2) overlapped with the consistently increased DEGs (n=1303); **b)** in RELA tumors, There was no gene overlapped between the signature genes and consistent decreased DEGs (n=281); and six genes (ARAP3, CYP27C1, GPSM1, JAG2, MCMBP and PCP4L1) overlapped between the signature genes and consistent decreased DEGs (n=208). To account for these overlapped genes, we have highlighted the overlapped genes the update Fig 1F (originally in Fig 1E and 1F).

7. In Fig3, the author claimed “the strong negative correlations between the DMRs and their target genes indicated a causative role of the DMR drivers in regulating the gene expressions”. This statement is overstated and not supported by experimental evidence.

Response: We agree and replaced “causative” to “potential regulatory”.

8. The definition of driver and booster is not solid enough to me, the conclusion inferred from them is therefore suspicious. Can the authors provide some reference or a third-party method to prove that the DMCs “that persisted in all the recurrences may have sustained (and boosted) tumor relapse and contributed to the increased tumorigenicity” indeed have a booster role?

Response: 1) We agree with the reviewer that, strictly speaking, a driver and booster should be validated by strong functional studies. Although we did not perform such functional study on single or selected genes, our data were generated from functionally (and clinically) validated tumor samples and provided a panel of genes (many of them, not all, novel in EPN) for detailed functional studies. 2) Since our data were derived

from clinically-proven recurrences, we performed analysis to identify consistent DNA methylation changes (candidate drivers and boosters) is to differentiate them from stochastic DNA methylation alterations. And, yes, we did find a similar approach to define/identify candidate drivers. Dan A Landau *et al* developed a statistical framework-MethSig to accounting stochastic DNA hypermethylation rate across the genome and between patients to infer DNA methylation drivers (PMID: 33972312), which required large number of samples. The advantage of our datasets is that we leveraged multiple recurrent EPN samples derived from the same patients to filter out stochastic DNA methylation alterations along relapse. Since tumorigenicity has been recognized as a useful/reliable assay in evaluating tumor malignancy, our identification of those DMCs persisted in the established PDOX tumors provided, at least partially, the functional evaluation of these candidate drivers/boosters, albeit this strategy has not been widely used due to high demand of time and effort. And, we do agree that these DMCs should be defined as possible or candidate boosters, and that is why we described them as “...*may have sustained (and boosted)*...”.

9. In Figure4C, could the author also show the biological enrichments of the DEGS that were negatively correlated with DNA methylation for RELA and PFA. What are the deep implications of these genes?

Response: This is something we tried, but our effort was limited by the small number of gene to yield a significant result in the enrichment analysis. At the same time, this relatively small gene list also provided us with a reasonable foothold for near future functional analysis (as therapeutic target or diagnostic marker) of the target genes in a manageable fashion. We addressed this in page 12:

Although the small number of dysregulated genes limited our capacity of detailed biological enrichment analysis, our discovery of their potentially new roles in promoting EPN relapses is exciting and warrants future functional validation and drug development.

10. In Fig5, the author reported several potential relapse predictors for in RELA and PFA tumors. As mentioned in the text, the sample size is relatively small. Then, how would the number of identified relapse predictors change as sample size changes? Would the number of candidates DMCs decrease as the sample size increases? If it's the case, how reliable the predictors are, since there could be large sample variation.

Response: This is a great question. Our effort of identifying relapse predictors is primarily driven by the clinical needs to prevent over-treatment of ependymomas (with unnecessary toxicity) that will not/or with low probabilities of relapse, or under-treated (to suffer from relapse) of tumors that will recur. The predictors we identified is aimed to provide a proof-of-principle to ignite additional efforts in the field of ependymoma studies so that large number of clinically annotated samples can be accumulated. And, yes, we do anticipate that the predictors will change as sample size increases. While it is difficult to predict if the DMCs will increase or decrease when the sample size increases (because it may increase the power to detect more DMCs), it is still highly desired that the number of candidate DMCs will decrease, especially following a progressively deeper understanding of the molecular subtypes of ependymomas and increased stringency. We added the following discussion in Page 16:

Despite sample variations of candidate predictors, it is highly desired that the number of predictors for EPN recurrence for each current or future molecular subtypes will decrease or be clinically applicable.

Minor concerns:

1. In Figure1c, it makes more sense to show the column dendrogram as well. In Figure1d, there are two RELA samples are closed to the Cerebrum samples, could the authors show the names of the two samples, and give an explanation of this situation? In Figure1EF, how the authors determine the signature genes?

Response: Yes, a column dendrogram is provided as Supplemental Fig 1D. As for the two samples in Fig 1D, the two RELA samples were RELA1-R3 and RELA3-R1. They did not form xenograft, and RELA1-R3 was obtained <4 months from RELA1-R2. Their closer proximity to the normal cerebral tissues (at DNA methylation) appears to suggest the contamination of peri-tumor normal brain/scar tissues in the surgical samples.

Supplemental Figure 1D. Column cluster dendrogram showing the relationships of the EPN tumors.

As for the signature genes In Figure1EF (also commented by Reviewer #1), they were extracted from published dataset GSE64415, which is primarily composed of primary tumors of ependymoma. The genes themselves were identified/defined by the authors of this database, and the levels of expression were from our dataset.

2. In figure1a, how to explain the correlation value decrease for the RELA2 between the 1st and 2nd relapsing?

Response: We believe the reviewer was referring to Figure 2A. We wish to thank the reviewer for pointing this out. It is interesting to see the decreased correlation value in RELA2, and that is why we pointed it out with an arrow. When these two relapsed tumors are compared, we can see from Figure 1A that RELA2-R1 was tumorigenic, but RELA2-R2 failed to form xenograft, which is different from other late recurrent tumors that displayed stronger tumorigenicity. While tumorigenicity can be affected by multiple factors, the reduction of PDOX formation from RELA R1 to R2 may indicate tumor tissue sampling differences (normal tissue contamination?) in addition to biological alterations.

3. Scale bar in Fig2B is missing.

Response: Yes, added.

4. In Figure2D, could the authors show the intersectional correlations between patient tumor and PDOX models? Such as the R7 RELA1 patient tumor to the R1 RELA4 PDOX model, et al.

Response: Yes, the intersectional matrix maps were presented as supplemental Figure 2D (Fig s2D).

Fig s2D. Matrix profile showing intersectional correlations of RELA (left) and PFA (right) tumors between patient tumors (*h*) from primary tumor (*P*) to the first recurrent (*R1*) and up to the 7th recurrent (*R7*) recurrences and PDOX models (*m*) up to passage 3 (*m3*) and 5 (*m5*).

5. could the authors give explain the deep implication of the number of DEG differences in the upset plots?

Response: We wish to thank the reviewer to point this out. We were excited about this comprehensive plot (that provided multi-parameters of important data) and added some more detailed description in the figure legend to better describe the graph:

A. *UpsetR plot showing the numbers of DEGs by comparing each RELA and PFA tumor tissues to normal cerebral and cerebellar tissues, respectively. The horizontal histogram represents the number of DEGs in each comparison between primary tumor and normal brain tissues; the vertical histogram represents the number of DEGs shared by tumors marked by connected dots. Arrows indicate the genes persisted from the primary to relapsed tumors. (Note of the label: P=primary tumor, R1= first recurrent tumor, R2=Second recurrent tumor, Dn=down regulated, Up=up-regulated).*

We also inserted the following sentences in the result section (page 10) to explain the deep implication of the DEG differences as below:

Comparison between PFA and RELA tumors revealed that the PFA tumors shared more DEGs (upregulated = 1,303 and down=1,290) and by 3 groups of tumors (from PFA-P to PFA-R1 and PFA-R20, which were remarkably higher than that in RELA tumors (up=relapses by three groups of tumors from PFA-P to PFA-R1 and PFA-R2 were significantly higher than those in RELA tumors (upregulated=208 and down-regulated =281) that were selectively shared by RELA-P and RELA-1 only (Fig S4A). The differences of cell-of-origin between RELA and PFA tumors^{20-23,39,55} may have contributed to the differences of DEG panels and the relatively conserved DEGs during PFA relapses.

6. Scale bar in Fig4A is missing.

Response: Yes, added.

Reviewer #2, expertise in ependymomas, genomics, DNA methylation (Remarks to the Author):

The study “Epigenetic Drivers and Boosters of Repeated Relapses in Patient-matched Childhood Ependymomas” by Zhao, Li, Zhang et al., presents an invaluable, rare, and well curated cohort of 30 ependymoma samples including patient-matched repeated relapses over 13 years. Similarly, the described cohort of 13 new ependymoma PDX models is of great value for future functional validation of new targets and treatments.

The authors rightfully focus on the question of why many ependymoma tumors recur. To uncover new molecular mechanisms of tumor recurrence, the authors decided to closely examine genome-wide DNA methylation patterns and generated a rich data set of gene expression and reduced representation bisulfite sequencing (RRBS). In line with previous reports, DNA methylation remains very stable throughout disease progression. It is really fascinating how stable the DNA methylation profiles are and even converge during repeated relapses. At the same time, this property somewhat limits the space for novel insights into ependymoma tumor biology by focusing on DNA methylation.

The authors have extensively used many different standard methods to analyze DNA methylation, such as the analysis of DMCs, DMRs, non-CpG methylation, and the association of DNA methylation patterns near

transcription start sites with gene expression changes. The results are a detailed encyclopedic review of DNA methylation patterns in primary and relapse PFA and RELA ependymoma tumors.

Despite the limited new insights into the molecular mechanisms that promote tumor recurrence and the lack of experimental validations, the study has a high impact as a resource due to the invaluable sample cohort and the epigenetic data obtained. Please find detailed comments below.

Response: We appreciate the insightful comments from the reviewer and we are very happy with support that the reviewer kindly provided.

1) "Driver" methylation events are defined, but there is no functional evidence of these regions' tumorigenicity. PDX models are presented as validation, but these models only show that the tumor is tumorigenic, not that DNA methylation causes that tumorigenicity. Functional validations are likely beyond the scope of the study, which is ok, but DNA methylation needs to be consistently described as a possible driver rather than a driver.

Response: Yes, we agree, and we have replaced "driver" to "possible driver" in the context (highlighted below). This is to supplement to the use of "candidate drivers" in multiple descriptions in our manuscript.

Page 9: These DMCs persisted from the primary tumors to late relapses, thereby constituting novel possible DNA methylation driver signatures of EPN relapse.

Page 10: This integrated analysis of patient tumors with their matching PDOX tumors represent a novel strategy for the discovery of functionally important target genes of possible DMR drivers.

.....To determine if the levels of these target gene expressions were actually regulated by the possible driver DMRs in EPN relapses

Page 13: Parallel analysis of these PDOX models also bridged a gap between the epigenetic reprogramming and the increased tumorigenicity by fine-tuning the newly identified DMC possible drivers and boosters critical for EPN relapses.

Page 14: In addition to discovering a whole new set of possible driver and booster genes for EPN relapses,

Page 15:and found a new set of possible DNA methylation drivers and boosters that sustained and promoted recurrence

2) I understand that abnormal DNA methylation present in primary tumors is defined by a comparison to the normal cerebellum/ cerebra tissue. This needs to be described more clearly in the main text.

Response: Yes, we described the normal tissues under Materials and Methods (page 15) and added the four tissues in Table 1. We also mentioned this comparison in Results (page 6)

3) A statistical problem is the treatment of dependent samples as independent. Eg., the authors claim to have found a set of methylation markers for RELA EPN relapse, but this set is dominated by markers specific to a single patient's multiple relapses. Given the rarity of EPN and their small sample size, perhaps this is unavoidable. However, these markers need to be tested on an independent dataset of EPN tumors.

Response: This is a very insightful comment. As suggested, we tested our potential relapse markers using dataset from GSE65362, which include 450k methylation array data for primary and 1st relapse EPN samples, and generated a map list of the overlapped CpG sites between our data and the 450k microarray data. It will be included as **supplemental Figure 6D**.

The follow description was inserted on page 12:

...we included an additional 4 PFA and 4 RELA primary tumors from a previous study (GSE87779) as a validation set (Fig. 4A, 4B, s6D) and identified a set of RELA and PFA recurrent specific DMCs in the relapse tumors in this public database (Fig. s6D).

Supplemental Figure 6D: Heatmaps showing the levels of a selected set of RELA (upper panel) and PFA (lower panel) recurrent specific CpGs' DNA methylations shared between our data (This study) and a previously published public data (GSE65362).

4) Page 6 “This finding is new and they demonstrated the maintenance of EPN molecular subtypes during repeated relapses (≥ 2) despite years of chemo- and/or radiation therapies (Fig 1A).” This is very impressive especially for the serial analysis. However, it has previously been shown that DNA methylation remains highly similar in the relapse compared to primary ependymomas.

Response: Yes, the previous study (GSE65362) showed that the 2nd relapse is highly similar to primary, and our study extended and beyond 2 relapses. Reviewer 1 also commented on the “this finding is new”. We have revised the statement as the following on page 6:

These set of data demonstrated the maintenance of EPN molecular subtypes during repeated relapses (≥ 2) during years of chemo- and/or radiation therapies (Fig. 1A).

5) Page 6 “For non-CpG methylation, the primary and recurrent tumors were found to have a dramatically decreased mCpA levels in both RELA and PFA tumors (Fig. S1F).” There is a striking difference of non-CpG methylation in the tumors as compared to the normal cerebellum/ cerebrum. Can the authors speculate what that means? Are these the same regions defined as abnormal DMCs/ DMRs?

Response: To answer the reviewer’s second question first, the CpAs are located in different genomic regions from the CpGs. As the reviewer nicely pointed out, our findings of the significantly decreased mCpA in both PFA and RELA tumors (at diagnosis and recurrences) was indeed very interesting. Although the differences between PFA and RELA tumors were not significant, our data seems to suggest that the decreased mCpA levels could be potential signatures of ependymoma. The following revisions were added under Results (page 6):

Non-CpG methylation is recently recognized as a novel layer of epigenetic information assembled at the root of vertebrates and plays new regulatory roles independent of the ancestral form of the CpG methylation⁴⁰. Its patterns are often tissue-specific⁴⁰⁻⁴³. In the present study, the primary and recurrent tumors were found to have a dramatically decreased mCpA levels in both RELA and PFA tumors (Fig. 1G) when compared with the normal tissues. Although the differences between RELA and PFA tumors were not

significant, this finding suggested that the reduction of mCpA can be a potentially novel epigenetic signature of EPN tumor of which the functional roles warrant further examination.

Under Discussion (Page 16):

Emerging data have suggested potential roles of non-CpG methylation in brain development and cancer biology^{40,41,43,81}. While our finding of the decreased mCpA in PFA and RELA EPN (regardless of relapses or not) was exciting, more cases and higher resolutions (e.g., whole genome methylation sequencing) are needed to draw definitive conclusions.

6) It is very impressive to see that late recurrent tumor(s) are much more likely to develop as PDX tumors. However, it's not evident that "convergent epigenetic reprogramming" promotes "tumorigenicity in EPN relapses". Gain of chromosome 1q has been identified as a genetic marker associated with recurrence and poor survival in PFA ependymoma tumors. For PFA ependymoma, the emergences of 1q gains might drive tumorigenicity instead of DNA methylation. Have the authors attempted to identify chr1q copy number status using the RRBS data and are there more frequent chr1q gains in the PFA relapses that grow out as PDX tumors?

Response: Yes, we are aware of the chromosome 1q gain ependymoma. Although it was not our primary goal to determine the role of chromosome 1q gain in the tumorigenicity of recurrent ependymomas, we agree with the reviewer that we should look into this issue in PFA. We applied CNVkit (<https://cnvkit.readthedocs.io/en/stable>) to infer CNV using RRBS data for PFA human and PDX samples, but we were not able to convincingly detect chr1q gain in our PFA human samples and PDOX tumors as shown in supplemental figure s3 (together with the description of methods). This is primarily because 1) RRBS sequencing reads are from enzyme cut regions, which results many identical reads, and 2) The enzyme digest is a random process that often led to uneven coverage within and among samples. These two limitations of RRBS data resulted in uncertainty in inferring CNVs. In fact, we have a hard time to find publications on CNV analysis using RRBS data, although the use of 450K array is well established.

To understand the chr1q copy number status, we tried a second approach by applying FISH to three patient tumors of PFA (from 2 patients) and their corresponding PDOX tumors (despite out extensive effort, the remaining PFA patient tumors were very difficult to obtain from Texas Children's. We detect chr1q gain in the patient and matching PDOX tumors, but the sample size was small and we could not get non-tumorigenic patient samples to compare. Therefore, more data are needed to definitively decide if 1q is a true driver of tumorigenicity. A series of revisions were made as detailed below:

In Results: We inserted the FISH results as Fig 2C, and added the following in Page 8.

Gain of chromosome 1q has been associated with recurrence and poor prognosis in PFA tumor^{1,50,54}. To examine if 1q gains might also be associated with tumorigenicity, we applied CNVkit to infer CNV status using our RRBS data. Due to the technique limitations, i.e., the identical reads and uneven coverage in tumors resulted from the enzyme digestion, there was not definitive

identification of 1q gain in PFA patient and PDOX tumors (Fig. s3). To supplement this approach, we applied FISH and detected 1q gain in paraffin sections of 3 sets of patient and xenograft tumors of PFA ependymoma (Fig 2C), suggesting that 1q gain in patient tumors were preserved in the PDOX tumors as well. These data support the analysis of additional PFA tumors, both tumorigenic and non-tumorigenic, to establish the role of 1q gain in PFA tumorigenicity.

In Discussion on page 17:

These set of data extended previous and our findings of chromosome 1q gain in promoting poor prognosis and potentially driving PFA tumorigenicity by providing a broader and higher resolution molecular signatures.

In Materials and Methods (page 17):

To infer DNA copy number status, particularly chromosome 1q, we applied CNVkit (<https://cnvkit.readthedocs.io/en/stable>) to infer CNV using RRBS data for PFA human and PDX samples. The CNVkits calculate normalized coverage in bin-level then it remove the systemic bias (such as CG content) use circular binary segmentation (CBS) to infer discrete copy number regions as segments.

Fig 2C. FISH analysis of chromosome 1q gain showing the locations of FISH probes in 1p (red) and 1q (green) (top), representative images of 1q (G: green) gain relative to 1p (R: red) (middle) and statistical analysis of matching pairs of patient (Pt) and PDOX tumors. ** $P < 0.01$

In Materials and Methods (page 19): **Fluorescence in situ hybridization (FISH) analysis:** *FISH analysis was performed as we described previously {Zhao, 2015 #13223} on 5 μm paraffin embedded sections slides using Vysis/Abbott Molecular (Des Plaines, IL) dual color probes targeting chromosome 1p36.3/TP73 and 1q25.2/ANGPTL loci, with the 1p36.3/TP73 locus labeled with spectrum red and the 1q25.2/ANGPTL labeled with spectrum green, for the detection of copy number alterations of both loci, following standard laboratory procedures in the clinical cytogenetics laboratory. A total of 100 non-overlapping cells were evaluated by two technologists independently. The average signals for both 1p36.3/TP73 and 1q25.2/ANGPTL were calculated. The signal ratio of 1q/1p ≥ 2.0 , is interpreted as gain or amplification of 1q25.2/ANGPTL.*

7) “These DMCs persisted from the primary tumors to late relapses, thereby constituting novel DNA methylation driver signatures of EPN relapse.” What we know is that these CpGs are consistently more/less methylated in EPN compared to normal tissue but that doesn’t mean those drive EPN relapse. It could be that whole cerebrum and whole cerebellum are in a different differentiation status than the actual EPN precursor cells and there might still be something completely different that drives tumor progression.

Response: Yes, we agree that it is possible the signatures can be associated with different differentiation status or developmental stage of the childhood brains. However, our patients aged from 2-10 at diagnosis and their relapses occurred from 1-13 years. Their normal brains (cerebral and cerebellar) have undergone significant changes. Therefore, we inserted the following sentences in Discussion on page 14:

However, childhood brains are often in different differentiation status, e.g., from 2-10 year old as in our cohorts. When combined with different time frame of recurrence, ranging from 1-13 years, some of the candidate DMC drivers may be attributed to the patient specific-differentiation status of cerebrum and cerebellum.

8) Fig.3 F: x- and y-axes need a separate legend for RELA (Primary or recurrent – cerebellum/ cerebrum)

Response: Completed as suggested.

9) “Most of the DMR regulated genes were newly discovered for EPN relapses. Their potential in as relapse driver genes was further enhanced by the fact that many of them, including CACNA1H, SLC12A7, CSPG4, RARA in RELA, and HSPB8, ITGB4, FAT1 in PFA tumors, have previously been associated with human cancers.” Many of these gene have previously been described in ependymoma, such as CACNA1H and WEE1, and some of them were experimentally validated as ependymoma tumor dependency genes.

Response: Yes, to reduce ambiguity, we added “aforementioned” to the first sentence. It now reads “Most of the *aforementioned* DMR regulated genes were newly discovered for EPN relapses.” We also revised the last sentence to reflect the role of CACNA1H as ependymoma tumor dependency“ *have previously been associated with human cancers or ependymoma tumor dependency gene (CACNA1H)^{61,62}*” The gene WEE1 was already mentioned and discussed in page 10.

10) “longitudinal analysis of consecutive, serially relapsing patient tumor samples will enable the separation of epigenetic driver(s) from transient random alterations”. Just because a mutation is conserved in subsequent relapses does not make it pathogenic; this would imply that non-pathogenic passenger mutations are somehow lost in subsequent relapses, which is not true.

Response: We believe the reviewer has made a very good point. However, this study is focused on DNA methylations, which unlike gene mutations, are dynamically modified under different biological and pathological conditions. We agree that transient alternations may not be “random”. We have thus deleted “random” from the sentence on Page 4 to convey this idea:

“longitudinal analysis of consecutive, serially relapsing patient tumor samples will enable the separation of *serially conserved candidate* epigenetic driver(s) from ~~random~~-transient alterations”

REVIEWERS' COMMENTS

Reviewer #1 (Remarks to the Author):

In the revised manuscript, the authors have addressed most of our concerns, however, there are still some questions that need to be further clarified.

1. For the RELA and PFA classification in Figure 1D, it seems the heterogeneity of RELA and PFA recurrent EPNs are derived from the brain region differences. The differentially methylated CpGs or expressed genes between RELA and PFA should be presented, and the relationship with the brain region specificity.
2. In Figure 1F, how the genes selected in the heatmap were not reflected in the previous main text, please revise accordingly and give a brief of how these genes were selected in the published database and how many of them were detected in your dataset.
3. In Figure 4C, at least for a few essential genes, the functional implication of the EPN replaces needed to discuss in the text.
4. For our major comment 10, we agree that the robustness of the identified potential predictors could be enhanced if more samples could be included in the datasets. As has been shown in the heatmap of Figure 5E, the methylation level at several CpG sites varies between different samples (different patients), such as chr17-80943940-80943942 and chr5-92910051-92910053, which suggests that there could be sample variations. However, the sample itself is invaluable precious, the revealed predictors may also be a great source for the field of ependymoma studies.
5. In Figure 2A for the major comment 4, the authors need to add the explanation in the text considering if no previous sample was available.

Reviewer #2 (Remarks to the Author):

Response to the revisions of the manuscript Epigenetic Drivers and Boosters of Repeated Relapses in Patient-matched Childhood Ependymomas by Zhao, Li, Zhang et. al.

The authors addressed all issues and improved them through extensive additional analysis and changes in the manuscript. I am specifically impressed by their attempt to analyze the chr1q status using additional computational methods and FISH imaging. The results appear to be in favor of prevalent chr1q gain as a genetic driver of tumorigenicity.

However, at this point we cannot rule out DNA methylation as an additional component promoting relapses and the authors have already diligently rephrased their manuscript to discuss DNA methylation as a possible/ potential driver. The only point remaining that overstates the conclusion of DNA methylation as a driver and booster of relapses is the title of the study:

Epigenetic Drivers and Boosters of Repeated Relapses in Patient-matched Childhood Ependymomas

I suggest something like Epigenetic Alterations in Repeated Relapses of Childhood Ependymomas as more appropriate.

I congratulate the authors for their great study and this important contribution to the field.